


# Dynamics of hydrological and geomorphological processes in evaporite karst at the eastern Dead Sea – a multidisciplinary study

Djamil Al-Halbouni[1], Robert A. Watson[2], Eoghan P. Holohan[2], Rena Meyer[3], Ulrich Polom[4], Fernando M. Dos Santos[5], Xavier Comas[6], Hussam Alrshdan[7,8], Charlotte M. Krawczyk[1,9] and Torsten Dahm[1,10]

[1]Physics of Earthquakes and Volcanoes, Helmholtz Centre - German Research Centre for Geosciences, Telegrafenberg, Potsdam 14473, Germany.
[2]Irish Centre for Research in Applied Geosciences (iCRAG), UCD School of Earth Sciences, University College Dublin, Belfield, Dublin 4, Ireland.
[3]Department of Geosciences and Natural Resource Management, University of Copenhagen, Øster Voldgade 10, Copenhagen 1350, Denmark.
[4]Department S1 - Seismics, Gravimetry, and Magnetics, Leibniz Institute for Applied Geophysics, Stilleweg 2, Hannover 30655, Germany.
[5]Instituto Dom Luís, University of Lisbon, Campo Grande Edifício C1, Lisbon 1749-016, Portugal.
[6]Department of Geosciences, Florida Atlantic University, 777 Glades Road, Boca Raton, FL 33431, USA
[7]MDA/IDC, Comprehensive Nuclear-Test-Ban Treaty Organization, Vienna International Centre, Vienna, Austria
[8]Ministry of Energy and Mineral Resources, Mahmoud Al Moussa Abaidat Street, Amman 140027, Jordan.
[9]Institute for Applied Geosciences, TU Berlin, Ernst-Reuter-Platz 1, Berlin 10587, Germany.
[10]Institute of Earth and Environmental Science-Earth Sciences, University of Potsdam, Karl-Liebknecht-Str. 24-25, Potsdam 14476, Germany

*Correspondence to*: Djamil Al-Halbouni (halbouni@gfz-potsdam.de)

## Abstract

Karst groundwater systems are characterised by the presence of multiple porosity types. Of these, subsurface conduits that facilitate concentrated, heterogeneous flow are challenging to resolve geologically and geophysically. This is especially the case in evaporite karst systems, such as those present on the shores of the Dead Sea, where rapid geomorphological changes are linked to a fall in base level by over 35 m since 1967. Here we combine field observations, remote sensing analysis, and multiple geophysical surveying methods (shear wave reflection seismics, electrical resistivity tomography [ERT], self-potential [SP] and ground penetrating radar [GPR]) to investigate the nature of subsurface groundwater flow and its interaction with hypersaline Dead Sea water on the rapidly retreating eastern shoreline, near Ghor Al-Haditha in Jordan. Remote-sensing data highlight links between the evolution of surface stream channels fed by groundwater springs and the development of surface subsidence patterns over a 25-year period. ERT and SP data from the head of one groundwater-fed channel adjacent the former lakeshore show anomalies that point to concentrated, multidirectional water flow in conduits located in the shallow subsurface (< 25 m depth). ERT surveys further inland show anomalies that are coincident with the axis of a major depression and that we interpret to represent subsurface water flow. Low-frequency GPR surveys reveal the limit between unsaturated and saturated zones (< 30 m depth) surrounding the main depression area. Shear wave seismic reflection data nearly 1 km further inland reveal buried paleochannels within alluvial fan deposits, which we interpret as pathways for groundwater flow from the main wadi in the area towards the springs feeding the surface streams. Finally, simulations of density-driven flow of hypersaline and under-saturated groundwaters in response to base level fall perform realistically if they include the generation of karst conduits near the shoreline. The combined approaches lead to a refined conceptual model of the hydrological and geomorphological processes developed at this part of the Dead Sea, whereby matrix-flow through the superficial aquifer inland transitions to conduit-flow nearer the shore where evaporite deposits are encountered. These conduits play a key role in the development of springs, stream channels and subsidence across the study area.



## 1    Introduction

Karst landscapes result from the dissolution by water of rocks or semi-consolidated sediments on and below the Earth's surface (Ford and Williams, 2007) . Typically, weakly acidic meteoric water, percolates into the subsurface, where it dissolves soluble material and enhances its porosity beyond initial matrix porosity and/or fracture porosity (Goldscheider, 2015; Hartmann et al., 2014; Price, 2013). Consequently, a characteristically well-evolved network of subsurface voids and conduits in karst groundwater systems enhances the hydraulic conductivity of the host materials to extremes rarely seen in non-karstified aquifers (Hartmann et al., 2014; Kaufmann and Braun, 2000).

The scale and geometry of karst conduit networks are variable, and they depend upon factors including the lithological characteristics of the host rock, the regional geological and climatic setting and the nature of hydrological recharge (Ford and Williams, 2007; Gutiérrez et al., 2014; Parise et al., 2018).  In limestone areas of high annual precipitation, for example, surface dissolution leads to the formation of enclosed depressions ('*solution dolines*'), which channel surface water into the underground  (Sauro, 2012; Waltham et al., 2005). Fractures formed along faults, joints and bedding planes within the host rock can be enlarged by dissolution to create branching networks of caves which are large enough to be entered by speleologists (Ford and Williams, 2007; Palmer, 1991, 2007, 2012). As a result of their global prevalence and ready access, karst systems formed in fractured limestone constitute the bulk of our present understanding of karst drainage systems.

In contrast, karst systems formed by dissolution of evaporites are less common and less studied. Evaporite deposits are commonly poorly-bedded and may form constituent parts of marine or lacustrine sedimentary deposits that are semi- to fully-consolidated (Warren, 2006). Due to the extreme solubility of evaporite minerals such as halite, they are only able to form in arid environments (Frumkin, 2013), such that surface recharge is limited as compared to typical limestone karst. Moreover, karst systems in young evaporites are characterized by a more dynamic, changing flow system due to the instability of the flow tubes (Ford and Williams, 2007). Base flow in these conduits may be of very low discharge, and surface flow may only occur rarely in ephemeral *wadis* (dry river valleys), where water often flows beneath the surface (Price, 2013; Salameh et al., 2018; US Geological Survey et al., 2013).

One of the most rapidly developing evaporite karst systems on the globe is found on the shores of the Dead Sea (Figure 1a). This hypersaline terminal lake, fed primarily by the River Jordan, lies within the ~150 km long and ~ 8 - 15 km wide Dead Sea basin (Garfunkel and Ben-Avraham, 1996). This basin has subsided rapidly from the late Pliocene to present (Ten Brink and Flores, 2012) due to motion along the left-lateral Dead Sea Transform fault (DSTF) system. During this time, several paleolakes of varying size and duration (Bartov, 2002; Torfstein et al., 2009) have existed within the basin. Since the 1960s, the modern Dead Sea level has declined from -395 m msl to -434 m msl (1967–2020; ISRAMAR, 2020), i.e. by 40 m as of 2020. The lake level fell by 0.5 m yr$^{-1}$ in the 1970's and by 1.1 m yr$^{-1}$ in the last decade. This has led to a dynamic reaction of the hydrogeological system and landscape around the lake shore (Kiro et al., 2008). New springs, stream channels, depressions, landslides and sinkholes have formed since the 1980s. These have developed both within alluvial fan sediments deposited by flash floods in *wadis* terminating at the lake and within mud and evaporite deposits of the former lakebed that have been revealed by the retreat of the shoreline (see e.g. Abelson et al., 2006, 2017; Al-Halbouni et al., 2017; Watson et al., 2019; Yechieli



et al., 2016 and references therein). The hazards posed by erosion and subsidence results in severe consequences for tourism, agriculture and infrastructure at the Dead Sea (Abou Karaki et al., 2019; Arkin and Gilat, 2000; Closson et al., 2009; Fiaschi et al., 2018).

There are several hypotheses regarding the nature of the subsurface hydrogeology at the Dead Sea and how this relates to the development of rapid erosion and subsidence around the shoreline. There is a consensus that the falling Dead Sea level increases the hydrological gradient adjacent to the lake. This increase promotes increased incision of existing stream channels around the lake shore, adjustments in the planform morphology of existing channels (e.g. increased sinuosity), and incision of new channels on the exposed lakebed (Bowman et al., 2007,
2010; Dente et al., 2017, 2019; Ben Moshe et al., 2008; Vachtman and Laronne, 2013). Several studies contend also that the base-level fall results in lateral migration of the fresh-saltwater interfaces in the subsurface, which enables relatively understaturated groundwater to invade evaporite deposits previously in equilibrium with Dead Sea brine, thus driving karstification of those deposits and subsidence of the ground surface (Salameh and El-Naser, 2000; Yechieli, 2000; Yechieli et al., 2006). Other studies have argued that karstification is driven primarily (or
additionally) by preferential upward flow of fresher groundwater into the evaporite-rich deposits via hydraulically-conductive regional tectonic faults (Abelson et al., 2003; Charrach, 2018; Closson et al., 2005; Closson and Abou Karaki, 2009; Shalev et al., 2006). Two main approaches have been used to simulate mathematically such hydrogeological scenarios in or near the DS rift valley: (1) a 'sharp-interface approximation' to the transition between hypersaline and fresh groundwaters (Kafri et al., 2007; Salameh and El-Naser, 2000; Yechieli, 2000) and
(2) a density-driven flow simulations that account for mixing of waters of contrasting salinities and densities (e.g. Shalev et al., 2006; Strey, 2014). Both approaches yield some success within the existing hydrogeological constraints.

There are also varying views of the form and make-up of the subsurface evaporite deposits, and varying emphasis
on the mechanisms of subsurface erosion that lead to surface subsidence. Many studies invoke chemical erosion (dissolution) of a massive salt-layer of up to 15 m thickness and lying at depths of 20 - 50 m. This layer is proposed to be dissolved by diffuse or fracture-controlled groundwater flow, which results in large intra-salt cavities that then collapse to produce sinkholes (Abelson et al., 2017; Ezersky and Frumkin, 2013; Shalev et al., 2006; Yechieli et al., 2016). Other studies emphasise physical erosion (*'piping'* or *'subrosion'*) of weakly-consolidated alluvial
sand and gravel deposits due to focussed, turbulent groundwater flow in such materials at depths of 5 - 20 m (Arkin and Gilat, 2000; Sawarieh et al., 2000; Taqieddin et al., 2000). Such studies further propose a combined physical and chemical erosion of thinly interbedded lacustrine salt, marl and clay deposits at the depths of 1 - 40 m, whereby salt dissolution and clay weakening by relatively fresh groundwater flow generates subsurface cavities and conduits (Al-Halbouni et al., 2017; Arkin and Gilat, 2000; Polom et al., 2018;). It has been shown that under the geologic
setting of the Eastern Dead Sea these void spaces are mechanically instable and may lead to different scales of morphological expressions of subsidence (Al-Halbouni et al., 2018, 2019, 2020). It is emphasised here that such varying views are not mutually exclusive (cf. Watson et al., 2019). Indeed some recent studies have emphasised linkages between the surface water flow, groundwater flow in a shallow conduit system and dissolution of deeper-





seated salt - particuary during flash flood events - as an important factor in sinkhole formation (Arav et al., 2020; Avni et al., 2016).

The use of indirect (non-invasive) methods to investigate subsurface hydrogeology in karst environments presents
unique methodological challenges due to the high degree of heterogeneity and discontinuity in the subsurface (Goldscheider, 2015; Goldscheider and Drew, 2007). An overview of indirect methods applied to the Dead Sea sinkhole phenomenon is given by Ezersky et al. (2017) and Salem (2020). Despite these challenges, near-surface geophysical methods have shown potential for cost-effective investigation of subsurface erosion in evaporite karst environments (see e.g. Abelson et al., 2018; Fabregat et al., 2017; Giampaolo et al., 2016; Gutiérrez et al., 2011;
Jardani et al., 2007; Krawczyk et al., 2012; Malehmir et al., 2016; Muzirafuti et al., 2020; Wust-Bloch and Joswig, 2006) and determining the actual configuration of the fresh-saline water interface (e.g. Kafri et al., 2008; Kafri and Goldman, 2005; Kruse et al., 2000; Meqbel et al., 2013). This is especially the case when geophysical surveys are supported by borehole investigations and other lines of evidence to aid interpretations. Geophysical investigations combined with borehole and hydrogeochemical data on the western shore of the Dead Sea, for instance, provide
strong evidence for dissolution of a massive salt layer in the shallow subsurface (as described above) with cavity formation as imaged by high geoelectric resistivities, and for linkage of this processes to sinkhole development in many areas there (cf. Ezersky, 2008; Ezersky et al., 2011; Yechieli et al., 2006). Such evidence is weaker or more ambivalent on the eastern shore, however (Al-Zoubi et al., 2007; Frumkin et al., 2011; Polom et al., 2018).

In this paper, we investigate the relationship between geomorphological change and subsurface groundwater flow on the Dead Sea's eastern shoreline, at a key part of the Ghor Al-Haditha sinkhole site in Jordan (Figure 1). We combine surface manifestations, remote sensing analysis, and multiple geophysical surveying methods of horizontally polarized shear wave ($S_H$) reflection seismics, electrical resistivity tomography (ERT), self-potential (SP) and ground penetrating radar (GPR). These are complemented by numerical simulations of the density-driven
groundwater flow.

First, we summarise the regional and local geology to aid interpretation of geophysical data and to assign realistic modelling conditions. We then use remote sensing and surface manifestations to describe the spatio-temporal evolution of a set of stream channels fed by groundwater to show a major shift in the local hydrology associated
with numerous sinkhole collapses, particularly at the head of a late-developed major channel formed in the exposed lacustrine deposits at the Dead Sea's eastern shoreline. We then present new ERT and SP datasets gathered at the head of this major channel to image the subsurface flow paths, as well as new ERT data that complements previous shear wave seismic survey data (Polom et al., 2018). We also present new shear wave seismic data further inland, near to the outlet of the main wadi terminating in the area. We provide constraint on the groundwater table by GPR
surveys. On the basis of all the above, we hypothesise that flow of subsurface groundwater undergoes a transition from weakly-focused matrix-flow near the main wadis to strongly-focussed conduit flow near to the shore before feeding the major channel. To test this, we present 2D density-driven groundwater flow modelling that suggests that such conduits are a necessary component of the groundwater system to generate realistic groundwater head and electrical conductivity conditions in the case of a regressing Dead Sea level. This multi-disciplinary analysis





leads us to present a refined picture of the groundwater system at Ghor Al-Haditha with improved insight into the links between surface subsidence and erosion and groundwater flow in the case of the regressing Dead Sea.

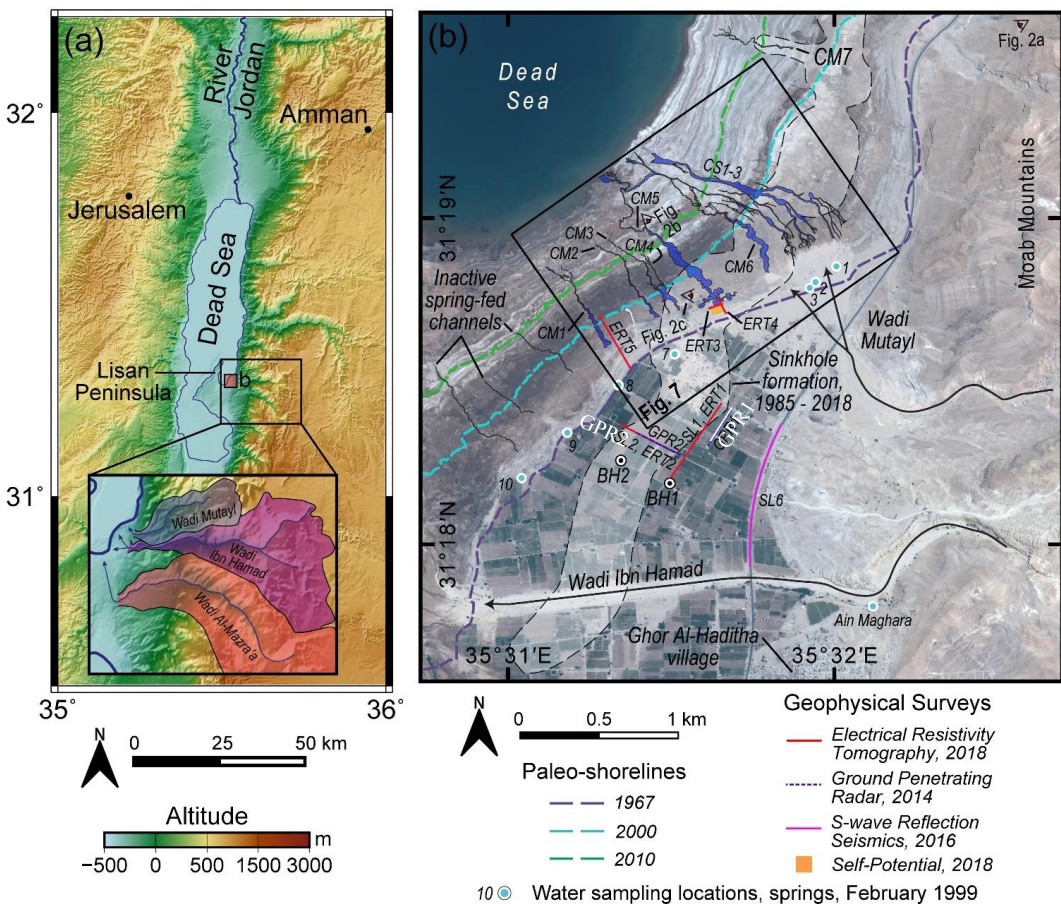

Figure 1: Overview of the Ghor Al-Haditha field site. (a) Topographic map showing the location of the study area and field site on the eastern Dead Sea shore, Jordan. Elevation data are derived from the Shuttle Radar Topography Mission (Farr et al., 2007). The inset shows the catchments of the three major wadis flowing to the Dead Sea shoreline in the study area: Wadi Mutayl; Wadi Ibn Hamad; and Wadi Al-Mazra'a. (b) Pleiades 2018 satellite image of the study area showing the studied groundwater-fed stream channels (blue) mapped from satellite imagery and close-range photogrammetric surveys. Label CM- refers to meandering type channels, CS- refers to straight or braided channels. Geophysical survey lines or areas, as analysed in this work, are marked. Seismic lines SL1 and SL2 refer to S-wave reflection profiles 1 and 2 acquired in 2014 (Polom et al., 2018). The two boreholes of El-Isa et al., 1995 are labelled "BH1" and "BH2". The area affected by sinkhole formation from 1985 – present is represented by the dashed outline with shaded infill. The numbered blue and white dots represent the approximate locations of groundwater springs sampled during a previous field campaign in the study area in 1999, from Sawarieh et al. (2000), and discussed further in Sec. 2.

## 2    Site of investigation and previous geophysical and hydrogeological studies there

The Ghor Al-Haditha study area is situated on the eastern shore of the Dead Sea (Figure 1). The climate is semi-arid to arid, with average annual precipitation of 70 - 100 mm (El-Isa et al., 1995; Salameh et al., 2018).Three major wadi systems, Wadi Ibn Hamad, Wadi Mutayl and Wadi Al-Mazra'a (or Wadi Al-Karak) terminating in the area have deposited sequences of semi-consolidated to unconsolidated alluvial fan deposits at the coastline (Figure 1b, Figure 2a). The drainage basins of the wadis are delineated in the inset of Figure 1. The catchment areas of the



wadis are: 30 km² for Wadi Mutayl; 124 km² for Wadi Ibn Hamad; and 188 km² for Wadi Al-Mazra'a. West and north of the alluvial plain, exposure of the former lakebed by the Dead Sea's recession has formed a 'mud-flat' or 'salt-flat'.

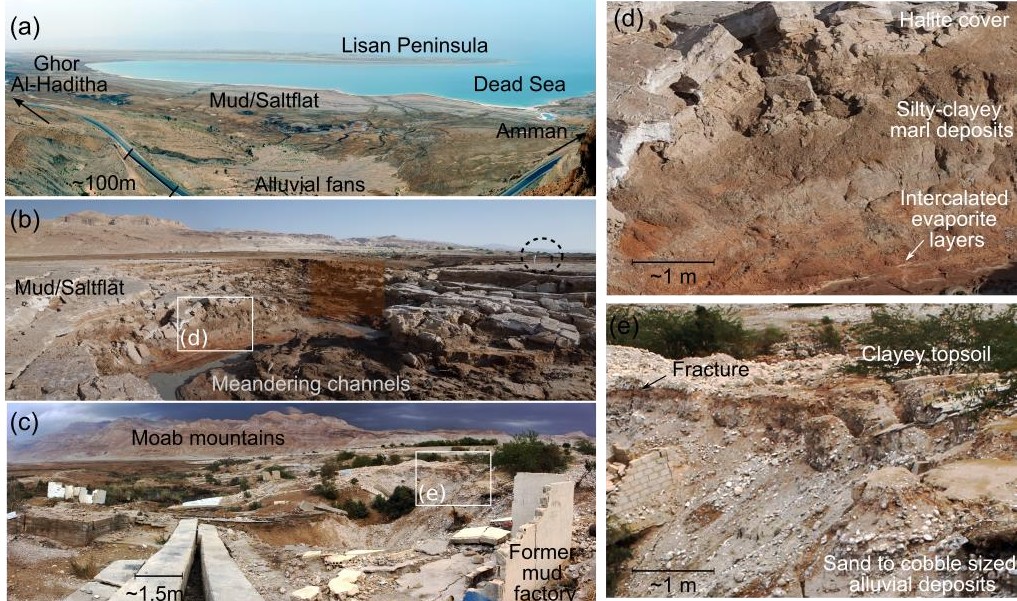

Figure 2: Field photos of the Ghor Al-Haditha study area. (a) View from the Moab mountains near Wadi Abu-Darat onto the former Dead Sea lakebed. (b) Vice-versa view from the DS lakebed towards the Moab mountains, with a section of a meandering stream channel and associated deformed ground. (c) Area around canyon CM5 with destroyed Numeira mud factory. (d) Zoom-in of lacustrine deposits. (e) Zoom-in of alluvial fan deposits.

The lacustrine deposits comprise alternating, light-dark, laminated to thin layers of carbonates (aragonite, calcite),
quartz and clay minerals (kaolinite), with cm-scale, idiomorphic halite crystals. These are interbedded with localised thin to thick beds of halite and/or gypsum (Figure 2b and d), the proportion and thicknesses of which near or at the surface increase lakeward and northward (Al-Halbouni et al., 2017; Watson et al., 2019). The alluvium consists generally of poorly-sorted, semi-consolidated to unconsolidated sands and gravels (Figure 2c and e), which are interbedded with minor silts and clays (El-Isa et al., 1995; Polom et al., 2018; Sawarieh et al., 2000; Taqieddin
et al., 2000). At the former stable shoreline, alluvium and lacustrine deposits are in direct lateral contact. Comparable deposits have been widely found along the western shore of the Dead Sea and in numerous boreholes there (Yechieli et al., 1993, 2002). Borehole data for the Ghor Al-Haditha study area are considerably less comprehensive, consisting of two boreholes, 45 and 51 m deep, drilled on the alluvial plain in January – February of 1995 (El-Isa et al., 1995; BH1 and BH2, Figure 1b, Figure 3). Borehole logs show alluvium down to the bottom
metre, where a layer of silt and greenish clay was detected. The exact ages of the alluvium and lacustrine deposits are unclear; the exposed lacustrine deposits likely correspond to the regional Ze'elim formation of Holocene age (Yechieli et al., 1993), while much of the alluvium may belong to the Lisan formation of Pliocene-Pleistocene age (Khalil, 1992).

There are three principal aquifer units in the area (Figure 3) whose characteristics are summed up in the following based on works of Akawwi et al., 2009; Goode, 2013; IOH and APC, 1995 and Khalil, 1992. The first is a lower





sandstone aquifer comprising the Ram group and Kurnub formation of Cambrian to early Cretaceous ages, respectively, which is not present at the surface in the study area but outcrops along the escarpment of the DSTF just to the north (cf. Figure 1c, Watson et al., 2019). The second is an upper carbonate aquifer spanning the Ajlun and Belqa groups of late Cretaceous to early Tertiary age. The third is a superficial aquifer in the Lisan and/or

Ze'elim formations. Recharge to the Ajlun-Belqa and superficial aquifers is primarily derived from precipitation in the highlands to the east, where the average annual precipitation is ~ 350 mm. As on the western side of the Dead Sea (Yechieli et al., 2006), it is estimated that some recharge occurs also via leakage from the lower Ram-Kurnub aquifer (cf. p5, Sect. 2.3, IOH and APC, 1995). From these regional aquifers, groundwater water flows toward the Dead Sea. A significant spring within the Ajlun group limestones, 'Ain Maghara', can be found just to the east of

the study area (Figure 3). The discharge of this spring is around 0.28 m$^3$ s$^{-1}$, and is thought to be principally derived (~ 80 %) from base flow within the Wadi Ibn Hamad (cf. p18, Sect. 5, IOH and APC, 1995), which in turn originates from outcrops of Ram-Kurnub aquifer unit along the wadi bed upstream of Ain Maghara. The remaining (~ 20 %) of flow is thought to derive from upward leakage from the Ram-Kurnub aquifer. Several springs are found in the transition between the alluvium and lacustrine deposits; these feed surface streams that drain into the Dead Sea via

numerous channels (Figure 1b, Figure 2b). Other surface stream channels in the lacustrine deposits lack these groundwater-fed springs and are instead fed by surface water from wadis during flash-flood events. A borehole drilled in the wadi bed just upstream of Ain Maghara confirms subsurface water flow within the wadi bed with a similar chemistry to that of the spring, though the depth and nature of this flow is not recorded. Head elevations in the vicinity of Ain Maghara are reported to be –300 to –350 m msl in 1994 (cf. pg 5, Sect. 2.2, IOH and APC,

1995). The depth to the water table in the superficial aquifer at the time of drilling in 1994 was 20.5 m in BH1; no depth was apparently recorded for BH2 (El-Isa et al., 1995). A groundwater well just over 5 km to the south of the study area, at the Al-Mazra'a pumping centre, indicates that the groundwater head level in the superficial aquifer declined at an average rate of 0.75 m yr$^{-1}$ from 1960 – 2010, whilst over the same time period the measured electrical conductivity had increased by 59 µS yr$^{-1}$ (US Geological Survey et al., 2013).

The first published report of geophysical investigations in the study area is El-Isa et al. (1995,) which includes results of classic refraction seismic and vertical electrical sounding (VES) surveys conducted in 1994. Several anomalies present in the refraction seismic data were inferred to represent buried alluvial channel deposits at 5–10 m depth, just southwest of BH2 (in the centre of their profile 4 and at the southern end of their profile 5; cf. Figs 4.9 and 4.11, El-Isa et al., 1995). Ten VES traverses were performed in the study area using an Atlas-Copco SAS-

300 system. The combined inversion of survey data across all traverses, calibrated with respect to the hydraulic head measured in the boreholes, enabled El-Isa et al. (1995) to create a layered 2D interpretation of groundwater conditions, consisting of an uppermost 'dry' area (resistivities of 120 – 660 Ωm), a 'fresh' groundwater layer (resistivities of 10 – 80 Ωm), a 'brackish' zone where groundwater and Dead Sea water mixing occurs (resistivities of 2 – 8 Ωm) and then saline Dead Sea water (resistivities of < 1 Ωm) and is sketched in Figure 3b. The water

levels in the boreholes were combined with spring elevation data to determine a hydraulic gradient of roughly 30 m km$^{-1}$ from east to west, following the surface topography.

A second report, by Sawarieh et al. (2000), includes additional refraction seismic surveys, as well as water chemistry, temperature, pH and electrical conductivity (EC) of springs, wells and sinkholes undertaken in February





1999 (see Figure 1b for the locations of the springs sampled). The seismic results generally resolved an up to 3-layer velocity model, thought to represent differing levels of compaction of alluvium. The hydrology of the study area has changed since the survey: several springs which they sampled in the area of Wadi Mutayl have now dried up, and the sampled locations do not match the locations of groundwater springs feeding channels CM1-3 and CM6

(Figure 1b). Groundwater samples were analysed for anion and cation concentrations and total dissolved solids (TDS) in the laboratory at the Ministry of Energy and Mineral Resources of Jordan (MEMR). The results are summarised in Table 1. In general, the measured TDS increased from east to west: groundwater sampled at Ain Maghara could be defined as 'fresh' (EC: < 0.75 mS cm$^{-1}$; TDS: < 500 ppm), whereas water rising from the springs closer to the shore would be categorised as 'brackish' (EC: 1.5 - 50 mS cm$^{-1}$; TDS: 500 - 35000 ppm), aside from

their sample 1 which had abnormally high EC and TDS, being more of a 'brine'. The Na/Cl molar ratios of water samples from these springs were relatively low. The EC readings from the field broadly correlate with the proportions of TDS and Na$^+$ and Cl$^-$ ions measured in the water samples.

Table 1: Water geochemistry of samples taken from groundwater springs during the 1999 field campaign of Sawarieh et al., 2000.

| Spring ID | Temperature (°C) | pH | EC (mS cm$^{-1}$) | Na (ppm) | Cl (ppm) | TDS (ppm) | Na/Cl (molar ratio) |
|---|---|---|---|---|---|---|---|
| 1 | 24.2 | 5.25 | 481.8 | 4700 | 194380 | 253260 | 0.04 |
| 2 | 24.0 | 6.60 | 33.8 | 3600 | 14683 | 22758 | 0.38 |
| 3 | 24.2 | 6.72 | 29.6 | 2550 | 12180 | 29245 | 0.32 |
| 7 | 24.3 | 7.05 | 8.3 | 910 | 3504 | 5787 | 0.40 |
| 8 | 24.3 | 7.00 | 13.3 | 1210 | 5339 | 8447 | 0.35 |
| 9 | 24.2 | 6.61 | 18.9 | 1510 | 6490 | 10684 | 0.36 |
| 10 | 24.2 | 7.05 | 12.6 | 860 | 4155 | 6684 | 0.32 |
| Ain Maghara | 24.2 | 7.16 | 0.9 | 100 | 192 | 663 | 0.80 |

From 2009 – 2010, scientists of MEMR performed geophysical surveys using ERT, GPR, and transient electromagnetic techniques (TEM, Alrshdan, 2012). GPR data acquired with a SIR-10B system near the mud-factory was interpreted as very shallow (< 7 m) sinkhole and subsurface water conduits, the same relates to surveys further inland. The used 100 MHz antenna and wet and saline ground conditions strongly restricted the penetration

depth of the GPR signal (Alrshdan, 2012). Similar GPR investigation of the shallow subsurface and sinkhole formation were reported by Batayneh et al., 2002. ERT surveys were performed traversing the alluvium-mudflat boundary close to CM1 and the former mud factory site, and further inland in farmers' fields parallel to the road along which ERT1 was performed in this study. The survey lines near the mud factory imaged some extremely high resistivity anomalies (> 10000 Ωm) thought to represent subsurface cavities. Three ERT profiles were

acquired with the IRIS Syscal system by Al-Zoubi et al., 2007, in the sinkhole area NE of Wadi Ibn Hamad. The higher resistivities up to 600 Ωm were interpreted as fractured alluvial material, while low resistivities of less than 20 Ωm were recorded at the bottom of all profiles. Two ERT profiles acquired further north in the study area by Frumkin et al. (2011) also imaged high resistivity anomalies which they interpreted to represent subsurface cavities.





All ERT and GPR surveys on the alluvial fan deposits only penetrated to maximum 25 m deep, which was insufficient to image the water table in this area.

The most recent geophysical survey of Polom et al. (2018), comprises a summary of shear wave reflection seismic data collected in 2013 – 2014. These data were interpreted as incompatible with a proposed 2 – 10 m massive salt

5 layer at around 35-40 m depth (Ezersky et al., 2013), but did reveal zones of high S-wave scattering in the subsurface near the main depression as well as potential buried and refilled channel systems within the alluvial fan architecture and a deeper lying "silt & clay" layer at around 60-80 m depth.

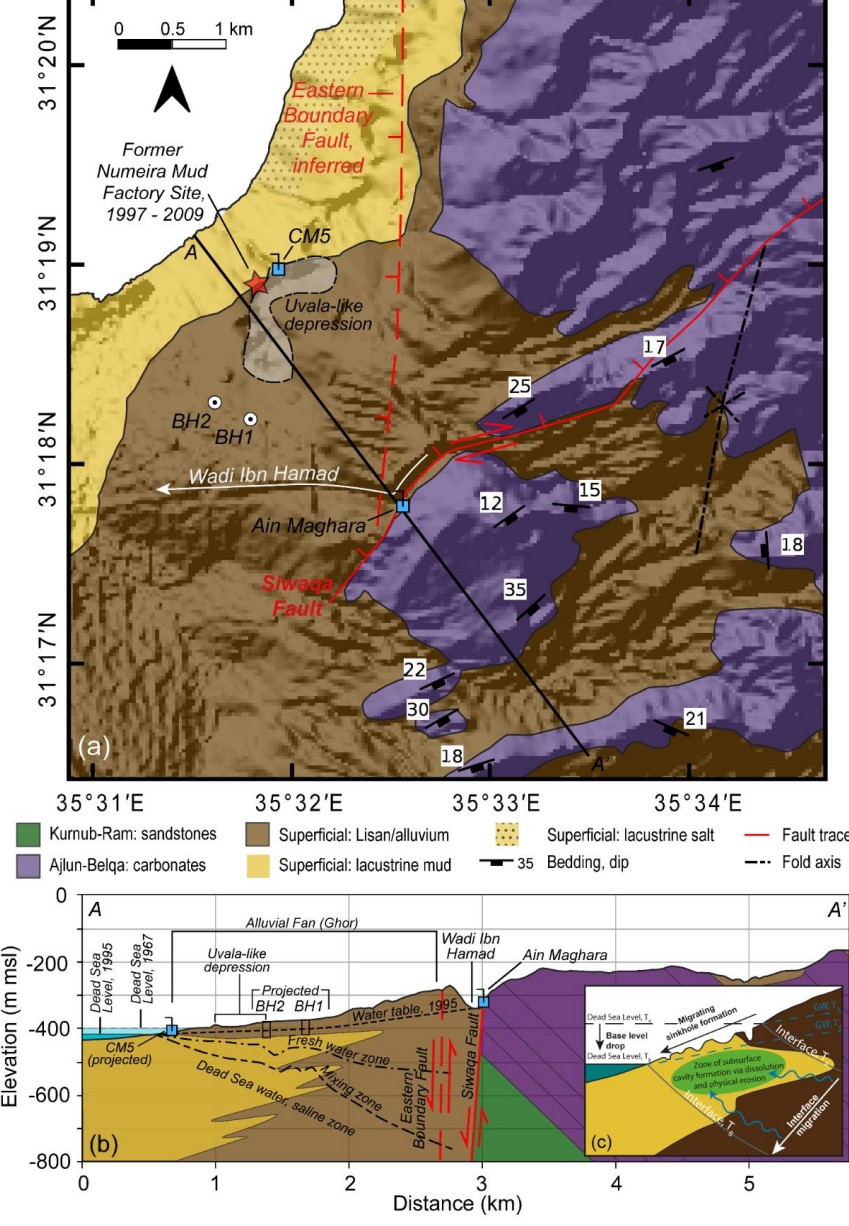

**Figure 3: Overview of the hydrogeology of the Ghor Al-Haditha study area. (a) Simplified geological map of the study area, partly**
10 **based on 1:50000 scale mapping of the Jordanian Ministry of Energy and Mineral Resources (Khalil, 1992), mapping and reports (IOH and APC, 1995) and partly on our own work. The stratigraphy generally dips acutely to the southeast, while striking to the**





northeast. Also shown is the right-lateral oblique Siwaqa fault, the inferred position of the Eastern Boundary Fault (downthrowing to the west), and the axis of the Haditha syncline. The present locations of the groundwater spring rising in the Wadi Sir limestones of the Ajlun group, 'Ain Maghara', and the spring rising at the contact between alluvium and mudflat deposits at the head of channel CM5 are also shown, along with the location of the large uvala-like depression described in (Al-Halbouni et al., 2017; Watson et al., 2019). A 2D cross-section of geology along the line A-A' is shown in (b). The red and black star indicates the position of the Numeira Mud Factory, which is defunct since 2009 after being destroyed by sinkhole formation at the site. (b) Cross-section of vertical exaggeration x5 depicting the subsurface hydrogeology as reported for the early months of 1995 (El-Isa et al., 1995), based on borehole information, refraction seismic surveys and VES surveys. The projected depth to the Ram-Kurnub sandstones is based on borehole data (IOH and APC, 1995). (c) Inset depicting the hypothesised conceptual model of seaward migration of sinkhole populations at the Dead Sea as the fresh–saline interface migrates with the base-level drop (Abelson et al., 2003, 2017; Salameh and El-Naser, 2000; Watson et al., 2019; Yechieli, 2000). „GW" here refers to the inferred level of the groundwater in the subsurface; squiggly arrows show the movement of water in the subsurface.

## 3    Material and Methods

Our multidisciplinary investigation of surface hydrology at Ghor Al-Haditha in Jordan consists of (1) remote sensing based on satellite imagery and aerial photogrammetric image analysis, (2) geophysical methods such as ERT, SP, GPR and $S_H$-wave reflection seismic surveys, and (3) 2D hydrogeological modelling of density-driven groundwater flow. Detailed theoretical concepts and additional results of the methods are illustrated in the Appendix A.

### 3.1    Remote sensing and Field surveys

The remote sensing dataset comprises high-resolution optical satellite imagery and aerial photographs that span the years 1967 to 2018. The dataset is identical to that presented in Table 1 of Watson et al. (2019), save for an additional Pleiades optical satellite image acquired on April 23rd, 2018. Temporally, the satellite and aerial imagery dataset is decadal in resolution from 1967 and annual in resolution from 2004, with spatial resolutions from 2 m pixel$^{-1}$ to 0.3 m pixel$^{-1}$. The remote sensing data is complemented by very high resolution (0.1 m pixel$^{-1}$) orthophotos and digital surface models derived by photogrammetric processing of optical images obtained through balloon- and drone-based close-range aerial surveys of the study area in 2014, 2015, and 2016 (cf. Table 1, Watson et al., 2019).

The satellite images were pre-processed (orthorectification, pansharpening and georeferencing and co-registration) by using the PCI Geomatica Orthoengine software package, as described in Watson et al. (2019), except for the 2016, 2017 and 2018 Pleiades images which were orthorectified and georeferenced by Airbus. For details of the method for generation of the orthophotos and DSMs from photogrammetric survey data see Al-Halbouni et al., 2017. After pre-processing, all remote sensing and photogrammetric datasets were integrated within the Q-GIS Geographical Information Systems (GIS) software package for geographical analysis, which included the manual digitisation of fluvial and karst features in order to reconstruct their development through both space and time.

Maps derived from remote sensing and photogrammetric data have been extensively ground-truthed over the course of four field campaigns in October 2014, October 2015, December 2016 and October 2018. The drone and balloon surveys were performed in conjunction with the first three field campaigns. In 2015, we also made a preliminary survey of water bodies in the study area. Temperature and electrical conductivity were measured in-situ at numerous springs and ponds (within sinkholes) with a Hach HQ40D Portable Multi-meter. These measurements were repeated for selected springs and ponds during the 2018 field campaign.





### 3.2 Geophysical Surveys

#### 3.2.1 Self-Potential

The Self-potential (SP) method is a passive geophysical method that measures the electric potential between two non-polarizing electrodes connected on the ground surface of the Earth. Although rarely used as a stand-alone

method, it is the only geophysical method able to resolve groundwater flow in many circumstances and on a large scale. It may be used to determine the depth of the unsaturated zone under certain conditions, and to locate drainage paths and flow tubes (Kaufmann and Romanov, 2016; Voytek et al., 2016). The main contributing physio-chemical mechanisms are of electrochemical, thermoelectric, telluric, and electrokinetic nature. The desired streaming potential signal (without disturbing effects) results from groundwater movement, an electrokinetic coupling

between groundwater and soil minerals, that creates the electric double layer (EDL, cf. e.g. Corwin, 1990; Overbeek, 1952; Revil et al., 1999a, 1999b). See Appendix A for further background information.

Our SP survey was conducted in 2018 (for location, see Figure 1b). We used the fixed base configuration (Figure 4a), with a high impedance (> 50 MΩ) Voltcraft voltmeter and several non-polarizable Ag/AgCl electrodes for the

survey (Figure 5e). The semi-permeable bottom filters of the electrodes are covered by a thick, wet bentonite mass inserted into a closed cotton bag. In this configuration, the electric potential is measured between a fixed base reference electrode, and a moving electrode. Adapted to our survey area, we measured parallel profiles to form an array configuration. The reference electrode, with the convention that the negative pole should be N or E of the positive pole, is buried and watered by using fresh water to improve electrical ground coupling. The moving

electrodes are also watered, and after the signal stabilized, we took at least three measurements for each point. More details on the SP array can be found in the results (Sec. 4.2.1).

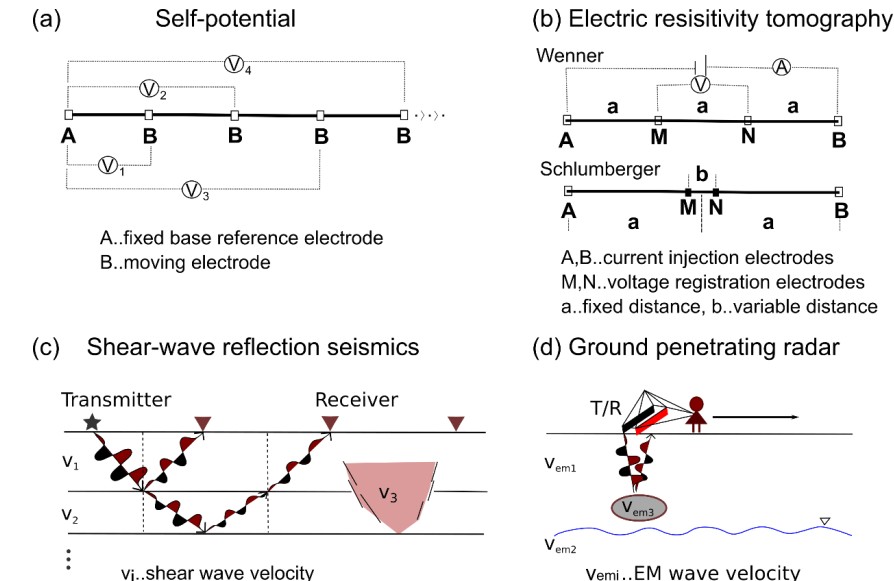

**Figure 4: Simplified sketches of the survey layouts used for the geophysical methods in this study. (a) Self-potential with fixed base**
**configuration applied to an array. (b) Electric resistivity tomography using the Wenner and Schlumberger electrode configuration layouts. (c) Shear wave reflection seismic principle using a geophone (elastic wave receiver) spread. (d) Ground penetrating radar principle along a profile.**



### 3.2.2 Electric Resistivity Tomography

Electric resistivity tomography is an active geophysical method within the family of geoelectric methods. Geoelectrics have been developed initially as vertical electric sounding (1D) to determine groundwater in the subsurface (Schlumberger, 1920) by injection and recording of an electric field using two pairs of electrodes. Nowadays, ERT works by injecting direct current (DC) or low frequency alternating current (AC, short time DC application) into the ground and measuring the differences of the electric potential field at multiple electrodes along a profile or array via computerized tomography (see e.g. Kirsch, 2006). The resulting apparent resistivity is based on Ohm's law, and the true resistivity of the ground is calculated thoroughly by 2D or 3D inversion (e.g. (Dahlin and Loke, 1998; Jardani et al., 2012; Loke et al., 2018; Loke and Dahlin, 2002). Small electrode spacing leads to high resolution and shallow penetration depths, whereas large electrode spacing lowers resolution and increases penetration depths. More details on these geometric considerations are given in the Appendix A.

Our ERT survey was performed in October 2018 by using a Lippmann 4point light 10W direct current instrument (Grinat et al., 2010) and stainless-steel electrodes connected to the Lippmann multichannel cable system (Figure 5b-d). Data were collected by using a multichannel control system implemented in the Geotest v. 2.46 software (Rauen, 2016) with automatic combination of all possible injection (AB) and measuring (MN) pairs of electrodes using a measurement interval frequency of 8 Hz. Different survey configurations for certain field conditions and penetration depths have been tested on site prior to the survey. Multichannel geoelelectric control systems commonly enable a combination of different general electrode configurations, i.e. Wenner, Schlumberger and several Dipole-Dipole configurations. The multichannel control system used here was a combination of Wenner and Schlumberger configurations (Figure 4b) which incorporates the advantages of both general configurations regarding advantages of lateral mapping and deep sounding. In profiles along the roads (ERT1 and ERT2) we used the so called roll-along technique to provide a combination of long profile distances (0 - 600 m) with short inter-electrode distances (1 - 5 m). Repeated measurements have been performed at certain electrode positions in case of non-reliable high resistance (> 1 KΩ) results and with support of fresh water to improve electrical ground coupling. Remaining anomalous data points (showing e.g. physically implausible negative resistivities or non-reliable local high resistivities) have been carefully removed from further analysis.

### 3.2.3 Shear wave reflection seismics

This method uses the principles of reflection seismology (Sheriff and Geldart, 1995), which is widely applied in hydrocarbon exploration. A seismic source at or close to the Earth's surface generates an elastic wave signal that propagates into the subsurface. The seismic wave is partly reflected at interfaces between elastically contrasting layers and then recorded by receivers (geophones) at the surface (Figure 4c). Another part of the wave energy is transmitted through the first layer interface and reflected and transmitted at the next layer interface. This continuous process enables the imaging of stacked subsurface layer structures with superior resolution and depth penetration compared to other geophysical methods. The main factors influencing the structure imaging are the wave propagation geometry, the propagation velocity and the signal propagation time from the source to the receiver. Shear body waves (S-waves) are not affected by pore space content (fluids, gases) and they propagate significantly slower compared to compressional body waves (P-waves). The resulting smaller wavelengths of S-waves can


improve the subsurface resolution by up to ten times. Furthermore, the S-waves propagate without any disturbance of groundwater, which is advantageous in alluvial sedimentary systems like the Wadi Ibn Hamad delta fan.

The shear wave seismic surveys reported on here used horizontally polarized S-waves ($S_H$-waves) and were

undertaken in October 2014 and December 2016. Details of the data acquisition and processing applied regarding profiles SL1 and SL2, which were acquired in 2014, are reported in Polom et al. (2018). Profile SL6 was acquired in 2016 using the same acquisition method and parameters. The profile was positioned perpendicular to the main structure dip of the Wadi Ibn Hamad alluvial fan with the intention of imaging a cross-section of probable fluviatile subsurface channel structures at the eastern boundary of the alluvial fan.

**Figure 5: Ground-based images of the ERT, SP and seismics equipment. (a) Seismics line 1 with landstreamer geophone system and ELVIS $S_H$ shear wave generator system. (b) ERT line 3 at the head of canyon CM5 with electrode chain ActEle spread. (c) ERT Lippmann 4point light and DGPS Trimble ProXRT-2 equipment. (d) Close-up of stainless-steel electrode, connecting cable and**

**switch box of the multielectrode ERT system. (e) SP Ag-AgCl electrode in a cotton bag filled with bentonite and watered to improve electric ground coupling. (f) GPR Malå Ramac device with 50 MHz transmitter and receiver antennas.**

### 3.2.4    Ground penetrating radar

Ground-penetrating radar uses the active emission of electromagnetic waves in the microwave band from a transmitter to a receiver antenna to investigate the relative dielectric permittivity of the shallow subsurface. The

method typically achieves vertical resolutions between 1-10s of centimetre depending on antenna frequency (Jol, 2009). The method has been widely used for decades in various sedimentological characterization studies (e.g. Neal, 2004) and has proven successful in identifying sinkholes and general analysis various attributes of karst systems, often in combination with other geophysical methods (e.g. Carbonel et al., 2014; Čeru and Gosar, 2019; Gómez-Ortiz and Martín-Crespo, 2012; Gutiérrez et al., 2011; Kaufmann et al., 2018; Kruse et al., 2006; Mount et

al., 2014; Sevil et al., 2017). The method has proven especially useful to determine the limits between unsaturated and saturated zone, image the water table in karst systems, and define the fresh-saltwater boundaries in coastal aquifers (e.g. Gustafsson, 2005; Igel et al., 2013; Kruse et al., 2000; Mount et al., 2015; Paz et al., 2017).





An electromagnetic impulse of a distinct frequency f is sent via a transmitter antenna into the ground, the ground response waves are recorded by a receiver antenna mounted usually close to the sender (so called vertical incidence case, Figure 4d). The survey is taken by dragging the ground coupled antennas along the surface with a steady movement to generate regular measurement intervals and setting digital marker points at distinct horizontal

distances to later match resulting trace numbers (individual measurements) versus horizontal distance.

All GPR surveys were conducted in October 2014 on common offset mode (transmitter and receiver antennas move simultaneously across a linear transect at a fixed distance) using a RAMAC GPR CU II system from Malå Geosciences (Figure 5f). Several transects were collected using both 100 MHz and 50 MHz unshielded antennas,

although only results from the later are reported here for a total of two transects (GPR 1 and GPR 2 in Figure 1b) that followed ERT surveys. Time-to-depth conversion for radargrams was based on estimates of EM wave velocities from three different approaches: 1) from diffractions in common offsets, yielding velocities ranging between 0.09 m ns$^{-1}$ for deeper materials to 0.15 m ns$^{-1}$ for surficial sediments; 2)  from field calibrations using a metal rod at a depth of 1.2 m buried on the wall of a collapsed sinkhole, with values ranging between 0.15- 0.2 m

ns$^{-1}$; and 3) from samples collected in the field at approximately 2 m deep in a sinkhole wall and measured in the lab under different moisture contents, reaching average values of 0.15 m ns$^{-1}$ for the driest (5% moisture content) conditions (Appendix A). The mean dielectric permittivity value hence lies around $\varepsilon_r \sim 4$, typical for dry sand.

Processing of common offset profiles was conducted with ReflexW v9.0 by Sandmeier Geophysical Research

(Sandmeier, 2019). Steps were limited to: 1) substract-mean (dewow); 2) background removal; 3) manual gain; 4) bandpass frequency; 5) Kirchhof migration (based on an average velocity of 0.15 m ns$^{-1}$); and 6) topographic correction. In some instances (i.e. GPR2 transect) the presence of isolated point reflectors (Neal, 2004) resulted in the presence of diffraction hyperbolas in CO profiles, allowing the construction of 2D models of velocity.

### 3.3    Inversion and Numerical modelling techniques

### 3.3.1    Inversion of SP and ERT data

In this study, all SP anomalies were assumed to be generated by a 2D polarized sheet that is inclined, thin and has an infinite extent perpendicular to the SP profile, in the following called "strips". Therefore, the model parameters are the position of the centre of the strip along the profile ($x_o$); its depth (h), the inclination angle ($\alpha_i$) and the half-width of the strip (a) and the polarization factor or electric dipole density (k). The Particle Swarm Optimization

(PSO) method was applied to determine values of those parameters (Monteiro Dos Santos, 2010). The PSO algorithm was first described by Eberhart and Kennedy (1995), it is a stochastic algorithm that simulates features of the social learning process as sharing information and evaluation of behaviours. The algorithm considers a community of N different models, and each model is updated taking into account its lowest RMS achieved so far (called the *'pbest model'*) until the best model by any model of the community (called the *'gbest model'*) is

obtained. The final solution will then be the gbest model achieved at the end of the iteration process.

The resistivity data were inverted by using the commercial program RES2DINV (GEOTOMO) v. 3.5 (Loke and Dahlin, 2002) that applies the Gauss-Newton inversion scheme. In a 2D model the Earth section below the acquired





profile is divided into numerous rectangular cells. The objective of the inversion procedure is to vary the resistivity value of each cell in order to find a collective resistivity response that best matches the apparent resistivity measured. The forward problem of the Jacobian Matrix value calculation is hereby addressed by a finite element simulation, that is, to calculate the apparent resistivity response of a specific resistivity distribution. A non-linear

L1 norm optimization method with smooth constraints is used to calculate the change in the resistivity of the model cells to minimize the difference between calculated and measured apparent resistivity (Loke et al., 2018; Sjödahl et al., 2008).

### 3.3.2 Hydrogeological modelling

A 2D hydrogeological forward model was developed by using Modflow v. 2005, Mt3DMS and Seawat v. 4 as

integrated in the FloPy environment (Bakker et al., 2016; Lee, 2018) to understand the effects of the falling Dead Sea level and of the hypothesised development of karst-related conduits on the groundwater system. With this model of falling base level, we simulated the evolution of groundwater level and salinity distribution in the superficial aquifer system along a 2 km long transect perpendicular to the shoreline of the Dead Sea (i.e. along a roughly Norwest-Southeast orientated profile following the NW section of profile AA' in Figure 3, with the centre

approximately at the head of canyon CM5). The model salinity distribution provided electrical resistivity values that were compared to those estimated from the ERT inversions. The 2D hydrogeological model setup and approach are presented in Figure 6.

The lithological units simulated in the model are: (1) alluvial fan sediments ('alluvium'), which comprise sand and

gravels, and which form a superficial aquifer; and (2) lacustrine sediments ('mud'), which comprise interbedded evaporites (such as gypsum, aragonite, calcite, and halite) and carbonates in clay to silt size (Khlaifat et al., 2010), and which is considered as an aquiclude (Abelson et al., 2006; Sachse, 2015; Siebert et al., 2014; Strey, 2014). Geologically, these lithological units represent respectively the terrestrial and lacustrine facies of the Pleistocene Lisan and/or Holocene Ze'elim formations. Alternating layers of alluvium and lacustrine sediments have been

simulated to take into account the local geology at Ghor al-Haditha, where a silty-clay layer lying under the alluvium is reported in boreholes and has been imaged seismically at 40-80 m below the surface at a distance of roughly 1 km from the ~1967 Dead Sea shoreline (El-Isa et al., 1995; Polom et al., 2018; Taqieddin et al., 2000). A similar geometric arrangement of aquifers and aquicludes is shown from boreholes for the western shore of the Dead Sea (cf. Yechieli et al., 2006).

The model dimensions are 2000 m by 160 m (Figure 6), and the model is discretized into a grid of 1 m wide by 0.5 m high cells. The Dead Sea (with a density of 1240 g l$^{-1}$, e.g. Siebert et al., 2014) is included towards the W. Hydraulic heads are calculated based on a reference datum defined at 525 m below mean sea level. No flow is assigned to the bottom of the aquifer system. Boundary conditions of constant head are defined at the Dead Sea

level in the NW, and a dropping water table by 0.75 m yr$^{-1}$ in the SE, according to the nearest available well measurements (cf. Sec. 2). The shrinkage of the Dead Sea is simulated as a yearly fall of the hydrological base level by 1 m vertically and as a yearly regression of the shoreline by 50 m horizontally for 18 years.

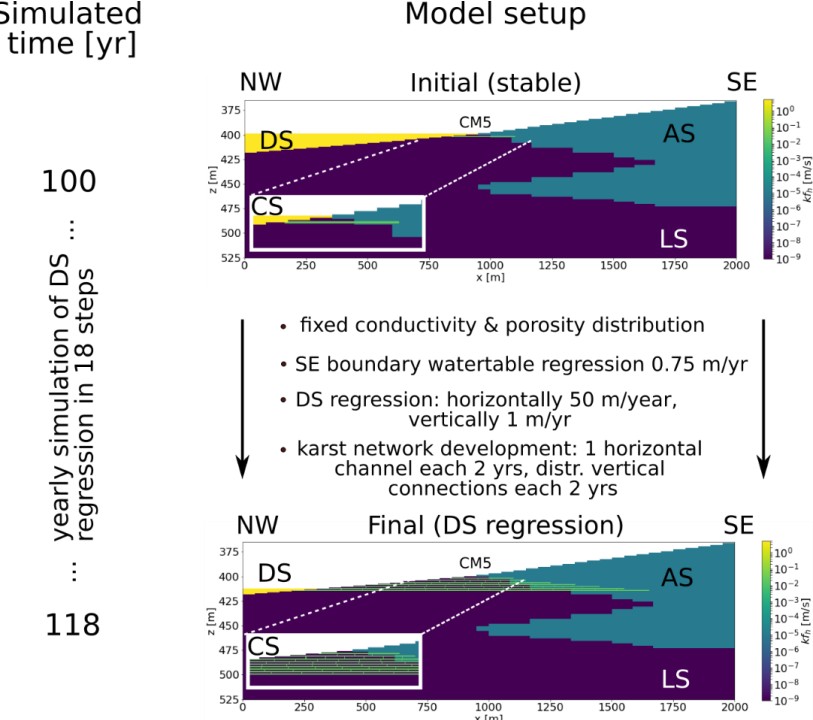

**Figure 6: 2D hydrogeological model and simulation time setup. The hydrogeological unit abbreviations stand for DS - Dead Sea, AS - alluvial sediments, LS - lacustrine sediments and CS - conduit system. Units are coloured by their horizontal hydraulic conductivity ($Kf_h$). The text gives a description of the modelling approach between the initial (top) and final state (bottom), after 18 years of**
**simulated DS regression and karst network evolution, which is presented as an inset. The top centre of the model corresponds roughly to the location of the outflow point of canyon CM5, the profile follows the NW section of the AA' line of Figure 3.**

The hydraulic properties in the model are mostly estimates in the absence of laboratory or in-situ data, and they are chosen to represent a simple aquifer/aquiclude system (Table 2). The horizontal hydraulic conductivities of $k_h = 1 * 10^{-6}$ m s$^{-1}$ for alluvium and $k_h = 1 * 10^{-10}$ m s$^{-1}$ for lacustrine sediments are taken as mean values from

literature as adapted for the DS in studies from Sachse (2015), Strey (2014) and Shalev et al. (2006). Vertical hydraulic conductivities are assumed to be 10 times less the horizontal ones, to represent anisotropy imparted from sedimentological layering. Hydraulic conductivities for the Dead Sea and for the hypothesised conduit system have been selected after various parameter tests. We added 1 m wide horizontal conduits of high effective porosity and hydraulic conductivity (Table 2), connected by 1 m wide vertical conduits, to the system in each second year of

simulation. Due to the groundwater level decline and shoreline recession, the horizontal extent of later-formed and deeper lying conduits is greater that of earlier-formed and shallower conduits. Conduit effective porosity and hydraulic conductivities were first varied in a trial & error approach to achieve model convergence and realistic steady state hydraulic results, as further explained below in Sec. 4.3.

We simulate density driven groundwater flow based on salt concentration in the porewater. The modelling approach is to first simulate the saline water distribution in an estimated steady state over hundred years before the Dead Sea started retreating, and to use this as initial condition for simulating the yearly lake recession. The starting salt concentration of 340 g/$^{-1}$ is adopted for the Dead Sea and for the lacustrine sediments, which are assumed to be initially saturated with Dead Sea water. A constant salt concentration is defined for the Dead Sea only. 'Salt' hereby





stands as a proxy for all types of dissolved evaporite minerals in the numerical modelling, with NaCl as the main component. The lacustrine sediments are assumed to be subject to a continuous fresh water inflow of initial salt concentration 0.67 g/$^{-1}$ through the alluvial sediments, adjusted to values of the Ain Maghara spring (Table 1). A hydraulic head gradient of ~ 30 m km$^{-1}$ (cf. Sec. 2; El-Isa et al., 1995; Sawarieh et al., 2000) is set, which is similar

5   to the topographic gradient in this area (Al-Halbouni et al., 2017). Basic diffusion and advection processes are added by using appropriate parameters for the different aquifer/aquiclude materials (Table 3). Convergence limits of the changes in hydraulic head between two iterations were set to 1/100, the same holds for changes in the concentration. Formation factors, effective porosities and the salt diffusion coefficient are known for the Dead Sea sediments analyzed samples of the western side of the lake (Ezersky and Frumkin, 2017; Yechieli and Ronen,

10   1996). Transport steps and dispersivities have been derived from different saltwater intrusion studies and the classical Henry problem (Abarca et al., 2005; Croucher and O'Sullivan, 1995; Geo-Slope, 2004; Langevin and Guo, 2006; Meyer et al., 2019; Zidane et al., 2012). For an overview of the initial spatial distribution of the effective porosity, concentration and head parameters please refer to the Appendix B.

Table 2: Hydrogeological model material parameters.

| Material | Hydraulic Conductivity, horizontal, kf$_h$ [m s$^{-1}$] | Hydraulic Conductivity, vertical, kf$_v$ [m s$^{-1}$] | Effective Porosity n | Formation factor f | Starting salt concentration C [g l$^{-1}$] |
|---|---|---|---|---|---|
| AS – Alluvium Sediments | 1.0e-5 | 1.0e-6 | 0.25 | 10 | 0.67 |
| LS – Lacustrine Sediments | 1.0e-9 | 1.0e-10 | 0.08 | 5.8 | 340 |
| DS – Dead Sea water | 5.0 | 0.5 | 1 | 1 | 340 |
| CS – Conduit System (Mixture) | 2.5e-2 | 2.5e-3 | 0.6 | 3 | 0.67 - 340 |

Table 3: Hydrogeological model simulation parameters.

| Model dimensions L/H [m] | Cell dimensions x/y [m] | Simulation length Perlen [yr] | Simulation transport step Dt$_0$ [s] | Diffusion coefficient D$_m$ [m² s$^{-1}$] | Longitudinal dispersivity α$_L$ [m] | H and V transversal dispersivity D$_{Th}$/D$_{Tv}$ [m] | Fresh/ Saltwater density w [g m$^{-3}$] |
|---|---|---|---|---|---|---|---|
| 2000/160 | 1/0.5 | 1-100 | 3.154e+3 | 1.89E-5 | 10 | 1/0.1 | 1000/1240 |



We estimate the bulk soil electric resistivities $\rho$ in our models from the salt concentration by using transformation equations between chloride concentration and TEM resistivities (Ezersky and Frumkin, 2017). Hereby we rely on water resistivity versus salinity relationships developed from borehole studies (Ezersky and Frumkin, 2017; Yechieli et al., 2001) and refer to the Total Dissolved Solids (TDS) content rather than the chloride concentration in our simulations.

$$TDS = \rho_W^{-1.229} \times 3.019 \text{ for low concentrations (TDS} < 131 \text{ g l}^{-1})$$
$$TDS = \rho_W^{-2.18} \times 0.247 \text{ for high concentrations (TDS } 131 - 320 \text{ g l}^{-1})$$

(1)

We can derive the bulk soil resistivities via $F = {\rho_w}/{\rho}$ by using Archie's law (Archie, 1942) even for the clay-containing DS sediments as they do not present cohesion and cation exchange effect after Ezersky and Frumkin (2017).

## 4 Results

### 4.1 Remote sensing and Field surveys

The new observations presented here regarding the geomorphological evolution of Ghor Al-Haditha are focussed on the area around the site of the former Numeira mud factory (Figure 7), which is coincident with the area in which the new geophysical surveys were conducted (see Figure 1b for overview). Here, stream channels of two distinct morphologies have been cut into the exposed evaporite-rich mud deposits of the former Dead Sea lake bed: meandering (CM) and straight (CS). The heads of all meandering channels have developed at spring points (in most cases, one per channel). Such springs lie either at the alluvium/mud-flat boundary or within the mud-flat deposits, and they originated at points that lie over 100 m seaward of - and several meters below - the 1967 lake level. The heads of straight channels do not correspond with spring points, but they have initially developed some distance out on the mud-flat, downslope from the termination of the active alluvial fan of the Wadi Mutayl.

As the shoreline has progressively retreated, both channel types have grown seaward, incising new channel segments into the lacustrine deposits of the former lakebed as the shoreline retreats over time. While the straight channels also show upstream growth (e.g. CS1-3 in Figure 7), most meandering channels show little or no upstream growth (e.g. CM1-4 and CM6 in Figure 7). Established sections of both channel types also widen progressively with time. From field observations, channel widening is commonly associated with fault-delimited slumping of the channel sides (Al-Halbouni et al., 2017). Both channel types deepen with time in response to the fall in base level; vertical incision is the primary fluvial erosive response to base level fall. The lower sections of the straight channels are commonly braided and contain deposits of sand to cobble clast size. Deposit grain sizes within the meandering channels are of mud to silt.

Sinkholes began developing in the area around the Numeira mud factory sometime after 1994, having first developed in the south of the study area in the mid-1980s (Sawarieh et al., 2000; Watson et al., 2019). By 2000, a





cluster of water-filled holes lay along the alluvium-lacustrine boundary to the north-east of the factory site, while another cluster had formed on the alluvium about 750 m to the south of the factory (Figure 7). Between 2008 and 2012, sinkhole development in this part of Ghor Al-Haditha accelerated dramatically and migrated from both of the initial clusters toward the factory site. In close spatio-temporal association with sinkhole development was the

formation of a gentle uvala-like depression (Figure 7). This depression formed on a scale much larger than the individual sinkholes, and much of its perimeter is delimited by numerous ground cracks and faults (Watson et al., 2019). By 2012, the section of this uvala along the boundary between alluvium and lacustrine sediments was occupied by a small lake.

A remarkable meandering channel, whose development highlights spatio-temporal links between subsidence and groundwater outflow, is CM5. This formed in the summer of 2012 with its head initially located in the middle of the mud-flat, about 750 m from the alluvium/mud-flat boundary (Figure 7). Headward channel incision progressed rapidly upstream over three months, in association with the drainage of the lake that had formed within the uvala (see Al-Halbouni et al., 2017, for details). Incision and headward erosion have continued at a slower rate up to

2018. The main springs feeding CM5 all lie near the centre of the uvala, at or near its deepest points. Since the establishment of CM5 in 2012, the growth of nearby meandering channels CM1-4 has since diminished markedly. Also, aside from one locality ~ 500 m to the SW, sinkhole development within the uvala since 2012 has been focussed within a 200 m radius of the head of CM5.

More detailed evidence of the dynamic development of erosional and subsidence features around the head of CM5 during the period 2014 - 2018 is shown in Figure 8. After its establishment in 2012 (Figure 7), channel CM5 was fed by a complex suite of springs whose activity has varied considerably between field campaigns in 2014-2018. In 2013, three springs, $s_0$, $s_1$ and $s_2$ fed the channel, with flow from $s_2$ rapidly eroding a new path to the main channel between June 2013 and September 2014. By the October 2015 field campaign, most of the flow into CM5 was

now provided by a new spring to the south, $s_4$. Active collapses of the $s_4$ stream head were observed over the course of a few days, between 25th - 28th October 2015, when a large rainfall event occurred, associated with flash floods locally elsewhere. These stream head collapses occurred in conjunction with sinkhole collapses along a line 20 - 30 m directly upslope of the stream head (Figure 8). Flow rates near to the channel head were around $2 \times 10^{-1}$ m$^3$ s$^{-1}$, and coarse sand to pebble-sized material was suspended in the flow (Figure 9e-f; cf. video supplement). The

other springs had either dried up or provided much reduced discharge to the channel. By December 2016, these sinkholes had been obliterated, as continued collapse and erosion produced a new section of channel with concurrent landward migration of spring $s_4$, which continued to be the main source of flow within the channel. The nature of $s_4$ was by now a proliferation of seepage points in the channel head (Figure 8). Discharge at one of these seeps was measured to be $2 \times 10^{-4}$ m$^3$ s$^{-1}$ in 2018, three orders of magnitude lower than during the high flow event

of 2015, and only very fine sand and silt particles were suspended in the flow (Figure 9d). This low discharge is also in line with qualitative observations of a low discharge rate at CM5 in 2014 and 2016 (as compared to the 2015 high flow event). Between December 2016 and October 2018, new water-filled sinkholes of considerable diameter (~ 30 m) had formed west of CM5. One of these holes was a new source of discharge to the canyon and is labelled $s_5$.





**Figure 7: Evolution of stream channels and karstic depressions from 2000 –2017. Left column shows aerial or satellite imagery. Right column shows maps of channels (red = straight/braided, green = meandering), ground cracks denoting the limits of a large-scale depression, and depression or sinkhole-hosted ponds (blue). Flow has converged upon CM5 in the years following 2012, with the other channels ceasing to prograde shoreward.**



**Figure 8: Evolution of the canyon-spring-sinkhole system close to the former mud-factory at the head of CM5. Left column shows aerial (balloon and drone, 2014 – 2016) or satellite imagery (2018). Right column shows maps of the channel (yellow), surface water (blue) and sinkholes (dark orange). Filled stars indicate active springs; empty stars indicate previously active springs. The size of the filled stars indicates the approximate relative contribution of that spring to the flow downstream of the T-junction between waters of the easterly and westerly springs.**





In-situ electrical conductivity measurements, as well as persistent localised vegetation growth around the ponds and in streams (Figure 8), indicate the discharge of brackish groundwater from these springs. Measurements of electrical conductivity made in October 2015 for each of the springs $s_1$-$s_4$ range from 13 - 22 mS cm$^{-1}$. Repeated in-situ conductivity measurements in November 2018 at $s_2$ and $s_4$ were slightly elevated as compared to 2015 but

show similar values of 26 and 22 mS cm$^{-1}$. For context, note that we measured electrical conductivity values of 100 - 220 mS cm$^{-1}$ in 2015 and 2018 at the springs (mostly sulphurous) emerging from the salt-dominated deposits further north within the Ghor Al-Haditha area (e.g. at the head of CM7 in Figure 1b).

Although limited direct evidence of subsurface groundwater channelisation has been observed, some subsurface

groundwater flow has been observed. At the shoreward termination of CM1, subsurface channels were observed in 2014 (Figure 9c) and have been observed for other channels to the north of the study area in Ghor Al-Haditha. These subsurface channels tend to be associated with more competent evaporite horizons within the lacustrine sediments, which appear to form the roofs of the conduits while the weaker clays are excavated beneath. Additionally, occasional 'pipes' have been noticed in the base of sinkhole collapses in both the alluvial gravels and

lacustrine mudflats (Figure 9a-b).

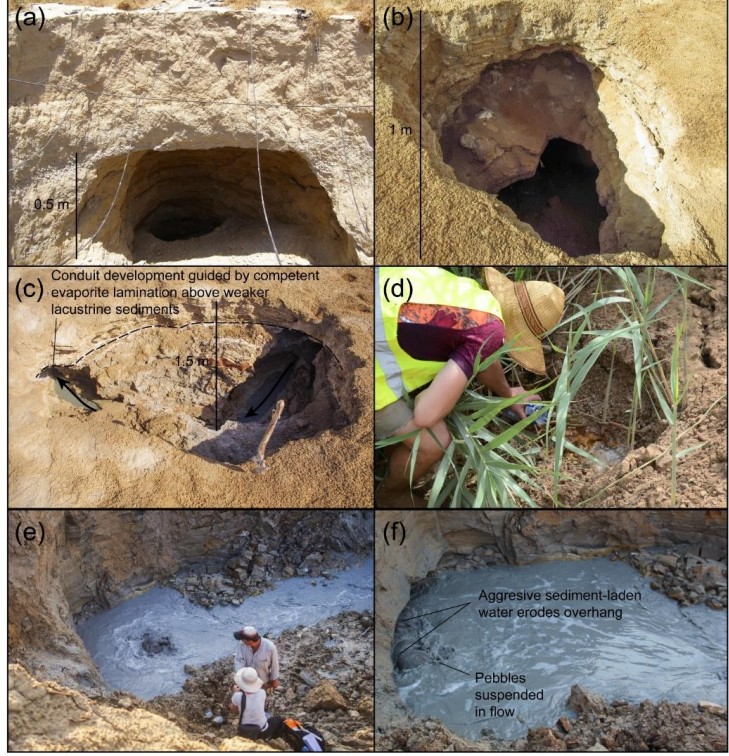

**Figure 9: Field impressions of conduits and karstic groundwater springs at Ghor Al-Haditha. (a) Nested sinkhole in the alluvium (located in the southern part of the uvala) intersecting a further cavity or pipe. Photo: Damien Closson, 2008. (b) Sinkhole in lacustrine mudflats (located just inland of CM7; photo taken 2018) with pipe disappearing toward the top of the picture. (c) Conduit**

**formed in lacustrine mudflat deposits near to the shoreward termination of CM1 (photo taken 2014). The flow has excavated a conduit in the weaker mud deposits below a more competent evaporite lamination. (d) Spring $s_4$ at CM5 at base flow rate (estimated discharge: $2\times10^{-4}$ m$^3$ s$^{-1}$). The main stream is fed by several similar seeps within the depression at the channel head. Photo taken: 2018. (e) $s_4$ during high flow three days after an intense rainfall event (estimated discharge: $2\times10^{-1}$ m$^3$ s$^{-1}$). Photo taken: 2015, modified after Al-Halbouni et al. (2017). (f) Close-up of $s_4$ during high flow. The nature of the sediment load varies from fine sediments to**

**pebble-sized clasts (up to 5 cm diameter). Photo taken: 2015, modified after Al-Halbouni et al. (2017).**



### 4.2 Geophysics

#### 4.2.1 ERT & SP results at the head of active spring-fed channel CM5

As demonstrated in the previous section, the area around the head of stream channel CM5 has been a highly active
zone of erosion and subsidence since its formation in 2012 (cf. Figure 8; Al-Halbouni et al., 2017; Watson et al.,

2019). These rapid geomorphological changes, occurring at the interface between alluvial cover and lacustrine
sediments, made this area an obvious target for geophysical investigation of possible subsurface structures (i.e.
cavities or conduits) in both materials. Consequently, near-surface geophysical surveys with the SP (array) and
ERT (lines 3 and 4) methods were performed here (Figure 1). The results of both surveys are shown in Figure 10.

Using the PSO 2dD inversion method (cf. Sec. 3.3.1), we extracted and inverted data for five profiles within
traversing the SP array (SP1-5, c.f. Figure 10a). The results and fit are shown in Figure 10d. The model misfit RMS
ranges from 5 to 15 %, while the SP data error has been estimated in the field to be 4.5 %. Due to the 2D nature of
the inversions, there is also a ~ 10 m horizontal error on the location of each anomaly (in x-y space, cf. Figure 10a).
From these inversions, we derived the dimensions and orientations of the inclined 'strips' of varying depths, widths

and inclinations which best represent the modelled curves. The location and extent of these strips, and the inferred
flow directions, is illustrated in Figure 10e. The length of each strip corresponds to the half-width of the matching
anomaly in Figure 10d. The direction of water flow along a strip to produce the inverted anomalies is generally
from minus (red) to plus (black). It should be emphasised that the modelled strips are by no means the only possible
'strip' arrangements able to explain each SP anomaly, particularly in regard to the depth of the modelled strips: a

long, deep strip may produce the same anomaly as a short, narrow strip. For clarity, the results of profile SP3 have
been moved to the Appendix A, which contains also the detailed geometric and electric parameters of each
inversion anomaly (Table A1).

The SP survey produced a complex dataset, the main anomalies of which are two large positive (red) patches with

a maximum potential of $\partial V = 50$ mV, where $\delta V = V - V_{ref}$ ($V_{ref}$ being the potential at the reference electrode).
These positive anomalies correspond to the pair of upflow anomalies (v) and (vi) along SP2, within alluvial gravel
deposits which have subsided and fractured considerably with the formation of the uvala there. The 'strip'
anomalies are proportional in magnitude to the size of each patch. Additional upflow anomalies (i) and (iii) are
present along SP1 with corresponding positive anomalies of $\partial V \sim 10$ mV in the array, and a horizontal flow

anomaly (ii). Strip (iii) results in a long-wavelength anomaly from – 10 to 10 mV in the second half of SP1. It
appears to be flowing toward the locations of the formerly active spring $s_3$. Other anomalies are recorded along
SP4 and SP5, though the signals here are much weaker and noisier given the error margin, hence the apparent
'upslope' flow directions of some of these anomalies have to be treated carefully.

ERT lines 3 and 4 offer overlapping coverage with SP1 and SP5 and generally show decreasing resistivity with
depth, stratified into two principal areas. The lower sections of both lines reveal an extremely low resistivity layer
($\rho_A = 1 - 5$ Ωm) beneath more resistive zones. The top of this extremely low resistivity layer lies at 8-10 m depth
beneath the centre of ERT3, and in the case of ERT4 it forms a 'wedge' in the north of the profile which is





coincident at the surface with the lacustrine deposits of the former lake bed in the final section of the profile. The upper, more resistive zones of ERT3 and ERT4 show a range of resistivities from low ($\rho_A = 5 - 30\ \Omega$m) indicated by light blue to light green colours, over middle ($\rho_A = 30 - 100\ \Omega$m) indicated by dark green to yellow colours, and high values ($\rho_A = 100 - 500\ \Omega$m) of reddish/violet colour. Patches of reddish/violet colours of very high

5 resistivity ($\rho_A > 500\ \Omega$m) occur where the surface is highly fractured, with significant ground cracks. Although there is little apparent correlation between the SP flow anomalies and any significant resistivity anomalies in the ERT data, all the SP strips are modelled at depths above the very low resistivity area within the upper, more resistive layers in a range of resistivity values between 15 and 100 $\Omega$m.

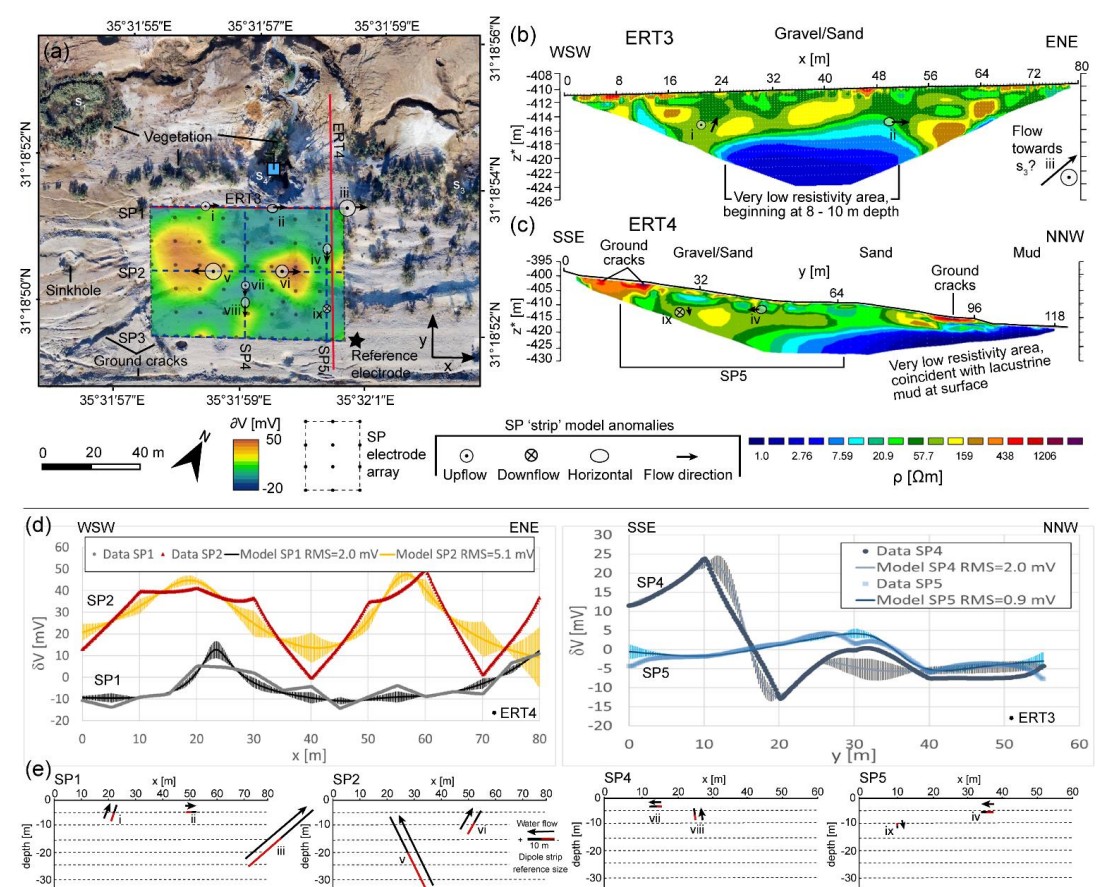

**Figure 10: Summary of SP and ERT survey results at the head of CM5. (a)** map showing the arrangement of the SP electrodes in an array and the resulting heatmap of potentials (with reference to a fixed electrode at the SE side of the array) overlain on orthophoto from December 2016. The locations of the inverted profiles SP1 – SP5 and their associated strip model anomalies are labelled, along
15 with the modelled flow directions for the anomalies. Also marked are the locations of springs $s_1$, $s_3$ and $s_4$. **(b)** Electric resistivity distribution along ERT profile 3, which used a Wenner configuration with an electrode spacing of 1 m. Data inversion gave an RMS error of 4.4 % after 8 iterations. The observed surface material and the locations and flow directions of SP strip anomalies are labelled. **(c)** ERT profile 4, which used a Wenner configuration with an electrode spacing of 2 m. Data inversion gave an RMS of 10 % after 5 iterations. The observed surface material and the locations and flow directions of SP strip anomalies are labelled. **(d)** SP
20 lines 1, 2, 4 and 5 data and derived best inversion model results with indication of RMS error. **(e)** Graphical representation of polarized strip anomalies matching the modelled SP line data. The direction of water flow along a strip to produce the inverted anomalies is generally from minus (red) to plus (black). Due to noise in the data, the exact peaks in the modelled self-potential curves are not always reached. All results from inversion of SP3 have been omitted from all figures for clarity; to see these results see Figure A1 in Appendix A.





### 4.2.2    ERT results at inactive dry spring-fed channel CM1

To investigate the similarities and contrasts between presently and formerly active groundwater-fed stream channels, we performed an ERT traverse which crosses the boundary between alluvium and lacustrine sediments next to the channel CM1, which was dry in 2018 (although it was weakly active during field campaigns in the years 2014-2016). The results of this survey are shown in Figure 11, which reveals, similar to ERT lines 3 and 4, an extremely low resistivity layer ($\rho_A = 1 - 5$ $\Omega$m) which again coincides with surficial lacustrine mud deposits. The ground above this layer in the south of the profile is more resistive but the spatial distribution of these resistivities is not uniform, with a range of resistivities from low ($\rho_A = 5 - 30$ $\Omega$m) indicated by light blue to light green colours, over middle ($\rho_A = 30 - 100$ $\Omega$m) indicated by dark green to yellow colours, and high values ($\rho_A = 100 - 500$ $\Omega$m) of orange colours. Locally, very high resistivity bodies ($\rho_A > 500$ $\Omega$m) appearing as patches of reddish/violet colours in the profile can also be observed. The extremely low resistivity layer is in horizontal contact with the more resistive zones in the centre (200 m) of the profile. At 150 m horizontal distance the depth to the low-resistivity layer is already 35 m. This zone of extremely low resistivity becomes more conductive towards the NW. Between 100 m and 140 m along the profile at depths below 10 m, there is a 'tongue' of more resistive (green) material, which links gradually to the conductor beneath at around 30 m depth.

Field observations of the surface deposits around the ERT profile indicate that the differences in lithological properties between the lacustrine mud and the alluvial gravels are responsible for their contrasting responses to the injection of electrical current. The lacustrine mud exposed in the nearby stream channel walls (Figure 11d) primarily comprises a light brown marl composed of carbonate and clay minerals, with some laminated horizons of evaporite deposits such as aragonite and gypsum (~ 0.5 cm thick, white-coloured laminations). Idiomorphic halite crystals of up to 1 cm diameter also occur throughout this material. In contrast, the alluvial gravels and conglomerates are  composed of mainly limestone and dolomite clasts with some basalt clasts (cf. El-Isa et al., 1995; Sawarieh et al., 2000), which are weakly to strongly cemented by minerals such as calcite, aragonite or gypsum. These minerals are dissolved and re-precipitated post-deposition as concretions on the clast surfaces and in the intervening pore spaces (cf. Bookman et al., 2004).

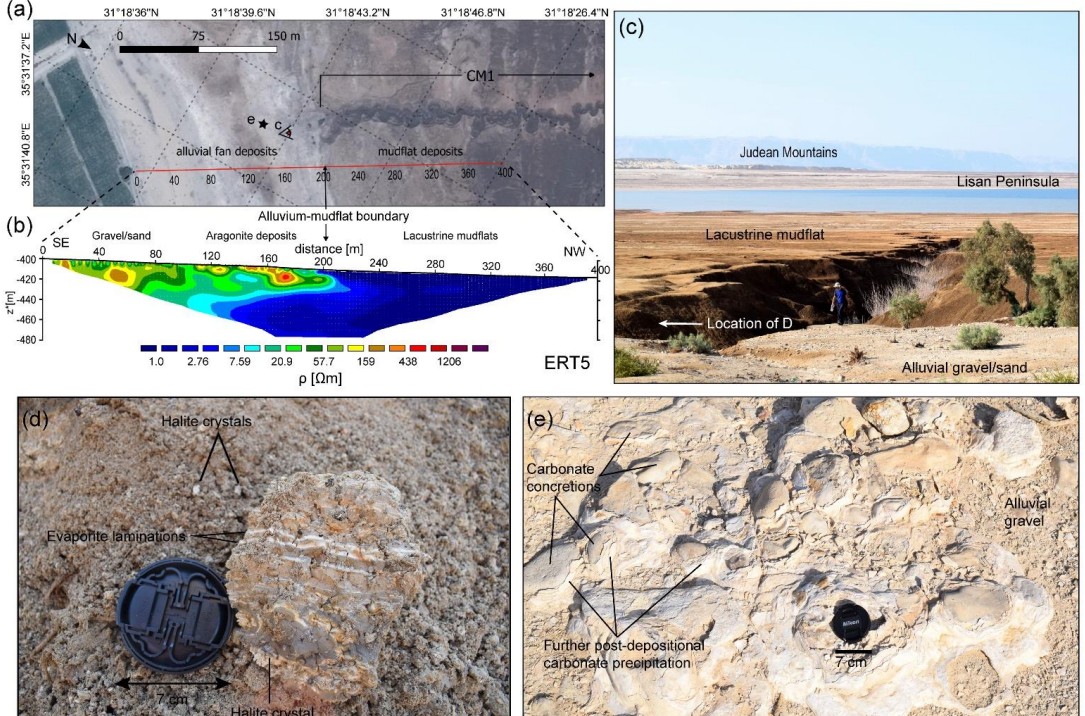

**Figure 11: ERT line 5 along the dry stream channel CM1 at the interface between alluvium and lacustrine sediments and associated surface manifestations. The survey used an inter-electrode distance of 5 m in a Wenner configuration. (a) Pleiades satellite image from April 2018 near canyon CM1 showing the profile location. (b) Electric resistivity distribution achieved by inversion of apparent**
**resistivities. An RMS of 7.5 % was achieved after 6 iterations. (c) Field photo from October 2018 looking NW along channel CM1, with contact between the alluvial fan deposits and the mudflat deposits of the former lakebed visible in the foreground near to the trees. Person for scale. The Lisan Peninsula and the Judean Mountains on the Dead Sea's western shore are visible in the background. (d) hand specimen of the former lakebed deposits from the bank of channel CM1, see part (c) for location. Lens cap for scale. (e) post-depositional carbonate concretions in the alluvial fan deposits, see part (a) for location. Lens cap for scale. The**
**minerals deposited are a mixture of aragonite, calcite, gypsum and clay minerals, all derived from the marls of the Lisan Formation. This degree of cementation is remarkable for the alluvium and is likely responsible for the elevated resistivity of the shallow subsurface at this location.**

### 4.2.3 Comparison of shear wave reflection seismics and ERT results in the main sinkhole area

To provide further context to the previous geophysical studies performed further inland in the study area (El-Isa et

al., 1995; Polom et al., 2018; Sawarieh et al., 2000), and to investigate possible changes to the subsurface hydrology

using electrical methods, we performed two new ERT surveys on the alluvial fan, at the southern edge of the uvala

(for locations see Figure 1). ERT lines 1 and 2 are coincident with seismic profile lines 1 and 2 as reported by

Polom et al., 2018, respectively (Figure 12a-d). The seismic profilings were carried out along asphalt-topped roads;

the ERT surveys were carried out on the soil next to these roads. Profile 1 transects the margin of a main uvala-like

depression in this area, while profile 2 crosses a subtler and narrower zone of subsidence that extends southwest

from the uvala. Numerous sinkholes formed adjacent to these profiles between 2002 and 2010, mainly within the

uvala and the narrow subsidence zone (Watson et al., 2019); those around profile 2 were filled in by local farmers.

Throughout the surveying time period (2014-2018), subsidence and related ground cracks (small faults or fissures)





remained apparent along both roads, but especially around the point where the road crosses the uvala at the northeastern end of ERT1 (Figure 12e-g).

In general, both ERT profiles are characterised by a vertical resistivity profile consisting of multiple distinct layers,
which agrees well with previous studies. A distinctive lobe with an uneven basal surface, defined across most of ERT2 and the southwest-central part of ERT1, is characterised by high ($\rho_A = 100 - 500 \; \Omega$m) resistivity values in the upper 30 - 40 m of the subsurface. This is underlain by a zone of low ($\rho_A = 5 - 30 \; \Omega$m) to extremely low ($\rho_A = 1 - 5 \; \Omega$m) resistivity values at subsurface depths below 40-50 m. The high resistivity layer is overlain by a thin ($5 - 10$ m thick) layer of moderate ($\rho_A = 30 - 100 \; \Omega$m) resistivity, corresponding to irrigated topsoil. This
apparent stratification is in broad agreement with the borehole results of El-Isa et al. (1995): the middle high-resistivity layer corresponds to interbedded sands and coarser gravels, below lower resistivities correspond to sand and even lower resistivities represent the silt and clay layers detected only at the base of the boreholes. The seismic data in these areas consist of reflectors dipping gently to the northwest, representing the topsets of the underlying alluvial fan system (Polom et al., 2018). Additionally, the resistivity values presented for ERT line 5 of Alrshdan
(2012), $\rho_A = 40 - 200 \; \Omega$m, are largely similar to those of the uppermost 25 m of ERT1.

A clear link between subsurface, low-resistivity anomalies in the ERT data and surface subsidence features is imaged in the northeastern section of ERT1, where the profile intersects the southern limits of the uvala. A striking area of reduced resistivities is apparent, with its centre around 120 m along-profile at around 20 m depth. These
low resistivities ($\rho_A < 10 \; \Omega$m) form a 'blob' around 50 m wide and 20 m high, with a rapid gradient at its edge to a surrounding area of greatly elevated ($\rho_A > 500 \; \Omega$m) resistivity. This region of elevated resistivity continues to the base of the profile here (Figure 12c). The corresponding part of the seismic section consists of strongly scattered, chaotic reflection patterns. The low-resistivity anomaly is directly beneath the area of maximum subsidence along the road, as shown in the aerial and field photos in Figure 12e-g. A smaller, vertically elongate low-resistivity
anomaly 50 m along ERT1 is also visible. This anomaly may be corroborated by ERT data from line 4 of Alrshdan (2012), which overlaps the northeastern end of ERT1 and shows a similar low-resistivity ($\rho_A < 10 \; \Omega$m) approximately 10 m diameter area at a similar depth ($< 10$ m).

Another significant anomaly in ERT1 is the strong horizontal resistivity gradient between the previously-mentioned
region of high ($\rho_A > 500 \; \Omega$m) resistivity to the northeast and a lower ($\rho_A = 10 - 20 \; \Omega$m) region to the southwest, around 250 m along-profile (Figure 12c). This linear feature is also visible in the seismic section: an offset of ~ 2 m between a strong reflector at ~ 80 m depth (just below the vertical extent of ERT1), downthrowing to the northeast, is marked by blue arrows around 260 m along-profile. The VES survey results of El-Isa et al. (1995) in this area also found evidence for a vertical disturbance of this nature, which they attribute to the presence of a fault.



**Figure 12: Comparison of shear wave reflection and ERT profiles. (a) Seismic line 1 (modified after Polom et al., 2018b). The inserted yellow trapezoidal area marks the shape of ERT1. (b) Seismic line 2 (modified after Polom et al., 2018b). The inserted yellow trapezoidal area marks the shape of ERT2. (c) Electric resistivity distribution along profile ERT1 achieved by an inversion of apparent resistivities in a Wenner-Schlumberger roll-along configuration with an RMS error of 11.9 % after 9 iterations. (d) Electric resistivity distribution along profile ERT2 achieved by an inversion of apparent resistivities in a Wenner-Schlumberger configuration with an RMS error of 3.8 % after 6 iterations. Borehole lithologies of BH1 and BH2 (projected) are derived from (El-Isa et al., 1995) and recoloured from (Polom et al., 2018) (e) Orthophoto image from 2014 of the damaged section of road crossing the uvala along ERT1. Six main fractures, F1 – F6, are identified. (f) field photo from 2014 of the subsided old asphalt road surface, which is offset on fractures F1 and F5 and locally overlain by aggregate. (g) field photo from 2014 of offset of the old asphalt road surface on fracture F6.**



### 4.2.4 Shear wave reflection seismics along a section of the Amman-Aqaba highway

The hydrology of the Wadi Ibn Hamad alluvial fan has evolved significantly since the inception of base level fall at the Dead Sea in the 1960s. This is highlighted in Figure 13a, which shows the 1970 configuration of the wadi's delta system. The presence of vegetation and water at the surface (represented by blue colours on the map) highlights numerous distributary channels for water emanating from the wadi, two of which diverge from a fork a few hundred metres from the fan apex and run from there toward the former Numeira mud factory site (Al-Halbouni et al., 2017). The north-eastern end of ERT1 appears to cross one of these old channels, near the low-resistivity anomaly detected in the subsurface there (cf. Figure 12c). After 1970, this surface water distributary system was significantly modified by the excavation of a single, straight exit channel for the Wadi Ibn Hamad, and by the development of agricultural fields in a grid-like arrangement (Figure 1b). An extensive area of dense vegetation directly at the former Dead Sea shore-line is nonetheless indicative of a sustained source of sub-surface fresh water around the Numeira mud factory site.

Seismic profile 6 runs along the Amman-Aqaba highway and crosses this former distributary channel close to its fork (Figure 13a). The seismic section in Figure 13b reveals a vertically extensive paleochannel system in the superficial aquifer as highlighted in Figure 13c, with the first interpreted horizons occurring at around 20 m depth and the deepest at around 120 m depth. The derived velocities between 240 ms$^{-1}$ and 600 ms$^{-1}$ for these alluvial gravels (Figure 13d) represent differing degrees of sediment compaction and/or cementation (cf. Polom et al., 2018). The axis of this interpreted palaeochannel system coincides with intersection of the 1970 distributary channel with the profile line. Synsedimentary faulting is visible to the north of the central axis of the paleochannel with downthrown to the southwest.



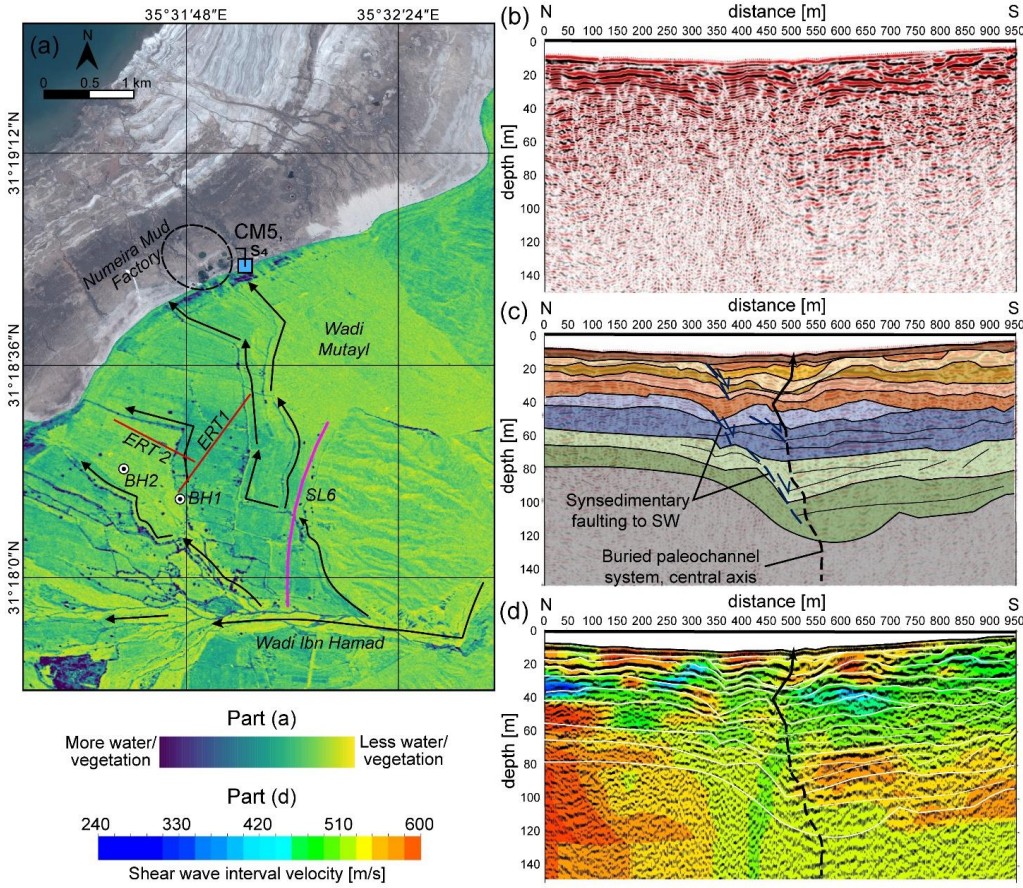

**Figure 13: Evidence of water flow in sediments of the Wadi Ibn Hamad alluvial fan as deduced from aerial photos and reflection seismic profile 6 connected to stream channel CM5. (a) Coloured Corona satellite image from 1970 overlain on 2018 Pleiades satellite image, depicting the former surface channel network in the alluvial fan deposits. Water content and vegetation appear in blue (modified after Al-Halbouni et al., 2017). Flow directions in the areas of elevated water content, as determined from the hydraulic gradient toward the Dead Sea, are highlighted by arrows to the right of the flow pathways. (b) Shear wave reflection seismic profile 6 along the Amman-Aqaba highway near Ghor Al-Haditha (cf. Figure 2). (c) Structural interpretation of seismic profile 6, analysis is described in Polom et al. (2018). Structures associated with development of a former channel system are recognized in the central part. (d) Shear wave interval velocity overlain on reflection profile 6.**

### 4.2.5    Water table inferred from GPR

GPR common offset profiles were processed according to Sec. 3.2.4 prioritizing the enhancement of deep reflectors in order infer the location of the water table during data collection in Oct. 2014 (Figure 14a and b). Both profiles show the presence of shallow (0.3 m ns$^{-1}$) diffractions indicative of air reflections and multiples that may result from ground coupling effects or ringing across the shallow most layer (due to the higher conductivity when compared to the shallow high resistivity zone below, Figure 12c and d). GPR profile 1 shows two major air reflections (i.e. hyperbolas, black arrows in Figure 14a) which correspond to buildings near the end of the line and near the gap in data (surface artificial concrete channel located between 190-210 m along the profile). In most parts of the profile, an undulating near-horizontal reflector between 400 and 450 ns is imaged (blue arrows), which corresponds to approximately 30 - 34 m depth when using an average EM wave velocity of 0.15 m ns$^{-1}$ as inferred from fittings to subsurface hyperbolic diffractions and lab tests. This reflector is interpreted as the limit between the unsaturated and saturated zone. The undulating nature of this reflector matches with the resistivity interface in





the corresponding parallel ERT1 survey (Figure 12c), despite being taken in 2018, that can be attributed to lateral contrasts in EM wave velocity. GPR common offset profile 2 (Figure 14b), corresponds to line ERT 2 (cf. Figure 1b), and also shows the presence of air reflections attributed to electrical posts placed parallel to the transect with consistent lateral spacing of 20 m intervals. Reflector facies for the first 8 - 10 m depth are again characterized by

apparent multiples (i.e. ringing) that seem consistent with ERT line 2 results showing the top 10 m high electric conductivity layer. Other diffractions (between 10 – 25 m depth) indicate signal velocities around 0.14 - 0.16 m ns$^{-1}$, consistent with very dry material. This zone corresponds to the high resistivity range found at similar shallow depth in ERT 2. There is the presence of a strong reflector (blue arrows) at around 500 ns two-way travel time (or ~ 38 m depth if using an average EM wave velocity of 0.15 m ns$^{-1}$. Fitting of hyperbolic diffractions in

GPR 2 are used to generate a 2D velocity model based on root-mean square velocities (Figure 14 c). The wave velocity shows a characteristic drop from 0.15-0.12 m ns$^{-1}$ to about 0.08-0.09 m ns$^{-1}$, at about 40-45 m depth that is consistent with a change in saturation and thus further supports interpretation of the deep reflector in Figure 14b as the water table. It appears more horizontal than in GPR 1, which may indicate the absence of strong lateral variations. When compared to the first 280 m of ERT 2, this depth corresponds to the transition zone between

moderate and low resistivity (10 – 100 Ωm).

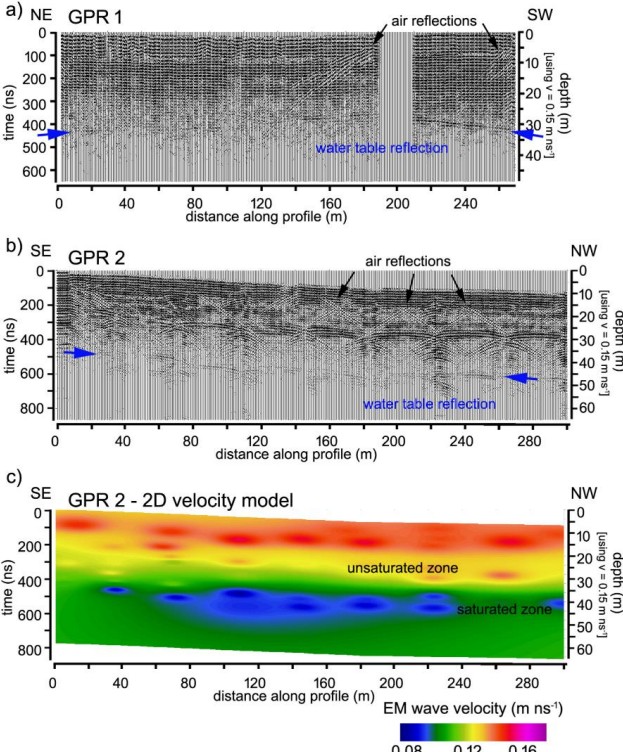

**Figure 14: Water table estimation from GPR. (a) GPR profile 1 running NE-SW parallel to ERT1 reveals a horizontal reflector between 400 and 500 ns two-way travel-time (~ 30 - 34 m depth below the surface). Note the gap between 190 and 210 m which is**

**due to a surface artificial concrete channel. (b) GPR profile 2 running NW-SE along line ERT 2 revealing a similar reflector at around 500 ns two-way travel-time (~ 38 m depth below the surface). Note that GPR profiles have been topography corrected via linear interpolation only, i.e. the depression is not shown in GPR1. Characteristic reflections have been marked by arrows. (c) GPR profile 2 electromagnetic wave velocity distribution. A characteristic drop in wave velocity is seen at the interface between saturated and unsaturated zone.**

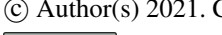



Furthermore, laboratory investigation of alluvial sample material with different gravimetric water content $\theta$ revealed a conductivity of 0.0027 - 0.45 S m$^{-1}$, i.e. a resistivity of $\rho_A = 2.2 - 370$ $\Omega$m for saturated ($\theta \sim 20$ %) and dry (in situ) alluvium ($\theta \sim 5$ %), respectively (53Appendix A).

### 4.3 Hydrogeological modelling

To study the effects of the falling Dead Sea level upon the subsurface hydrogeology in the study area, we applied the finite-difference groundwater flow and transport model including density driven flow from the MODFLOW-family, MODFLOW2005, MT3DMS and SEAWAT integrated within a FloPy environment. We performed distributed simulation modelling of density-driven flow of the hypersaline Dead Sea water and fresh groundwater undergoing base level drop of 1 m yr$^{-1}$ for 18 years, after 100 years of hydrogeological steady state conditions. The

presence of subsurface conduits is simulated in the model by assigning increased porosities to linear arrangements of cells (an equivalent porous medium), with new conduits added at each timestep. In our model-space, realistic simulations (as compared to studies on the Western shore and elsewhere) of the evolution of salt concentration and groundwater head under density-driven flow can only be achieved by including these simulated conduits in the model space.

The main criteria for evaluating the quality of the hydrogeological model solutions were: (1) model convergence; (2) realistic hydraulic head development without singularities or jumps, which is expressed as a smooth water table as the model converges. Modelling the contact between alluvial and lacustrine sediments as an aquifer-aquiclude boundary with highly contrasting hydraulic conductivities did not yield model convergence. Only a drastic and

unrealistic increase of the bulk hydraulic conductivity of the lacustrine sediments – to values greater than the alluvium - gave a realistic hydraulic head (see Appendix B). On the other hand, a local increase of the hydraulic conductivities in both alluvium and lacustrine sediments - i.e. the inclusion of the simulated conduit network - achieves both convergent solutions and realistic hydraulic heads. The final model results incorporating such a conduit network are shown in Figure 15.

With the regression of the Dead Sea, the hydraulic heads fall, and the water table declines (Figure 15a), initially at the shoreline and then propagating further inland. This causes a strong gradient in the hydraulic head and water table that shifts further upslope during the regression. The groundwater level consequently decreases in the top 40 m of alluvial cover in a short time (10 years). At a location of x = 1500 m, the resulting head drop during the

simulated 15 years estimates to ~ 20 m.

After the initiation period simulation of 100 years under steady state conditions, the salinity of the system (Figure 15b) is initially in a quasi-equilibrium with a clear distinction of fresh and saline water according to the local geology. The fresh-saline water system reacts relatively slowly to base level fall because the diffusion of the saline water is a slow process. As such, the reaction of the hydrogeological system is visible only after 15 years (or 15 m

vertical lowering of the DS). The acceleration is caused by the development of a network of conduits. Formerly saturated areas become dry, and a rather salty water composition dominates the flow in the conduit system with concentrations between 125 and 250 g l$^{-1}$. The fresh-saline interface system builds locally a complex shape, and





different levels of saline water can be encountered in a hypothetical vertical slice, but also lateral variations appear. Regionally it evolves into a landward inclined shape of a Ghyben-Herzberg style interface.

The soil resistivity shows a duality of responses as the model develops, increasing in the near-surface lacustrine

sediments (due to renewed inflow of fresher groundwater in the conduits), but decreasing gradually (Figure 15c) at the diffusion fronts. The latter is especially true for the alluvium-mud boundary, but also in the implemented channel water system ranging far inwards the alluvial sediments. Due to saline water intrusion inland, the resistivity is gradually lowering in the deeper areas of the model, including areas of alluvium that were initially saturated by fresh water (US Geological Survey et al., 2013).

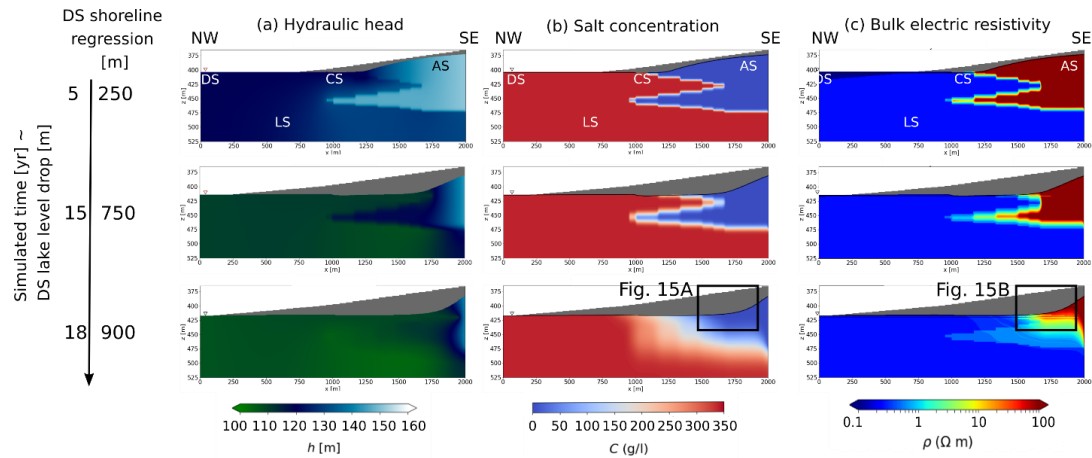

**Figure 15: Hydrogeological final model results incorporating a simulated conduit network. (a) Water head distribution. (b) Salt concentration and (c) Bulk soil electric resistivity. Different Dead Sea regression phases with an indication of vertical drop and approximate regression at this bathymetry. LS stands for lacustrine sediments; AS are the alluvium sediments; DS is the Dead Sea**

**and CS stands for the conduit system. The grey areas correspond to the unsaturated zone – i.e. the area above the water table.**

At the contact between the alluvial and lacustrine sediments (at a distance of 1500 and 1900 meters from the shore; Figure 16), which also corresponds to the site of the uvala-like depression formed around CM5, a complex, multidirectional flow network is observed to form in the latter years. The flow is initially horizontal and evolves into a mixture of horizontal and vertical flow during the regression of the DS (Figure 16a). Locally, upflow can

occur with velocities up to ~ 1e$^{-6}$ m s$^{-1}$. At the last stage, strong preferential flow of ~ 1e$^{-4}$ m s$^{-1}$ is dominated in a main conduit. Above the conduit, upward and downward flow patterns evolve of ~ 2e$^{-7}$ m s$^{-1}$, below the conduit infiltration dominates.

For comparison with resistivity structures observed in our ERT surveys, Figure 16b shows the slice of the simulated

bulk soil resistivity results for saturated areas. We can see a typical evolution at the contact zone between very conductive lower layers and very resistive upper layers. Once the saline water intrusion has developed further, the resistivities are modulated gradually depending on salt concentration and material type. The conduits appear usually as more conductive lines in both material types.





Important features of the numerical modelling that may assist in interpreting the ERT inversion results are as follows:

 a) Decrease of the alluvial fan resistivities from 65 $\Omega m$ to lower than 5 $\Omega m$ when subject to saline and highly saline water saturation.

 b) Very low resistivities of highly porous lacustrine/alluvium sediment channels of 0.5 - 5 $\Omega m$.

 c) Increase of the previously brine saturated lacustrine sediment resistivities from 0.1 $\Omega m$ up to 1 $\Omega m$ due to refreshening.

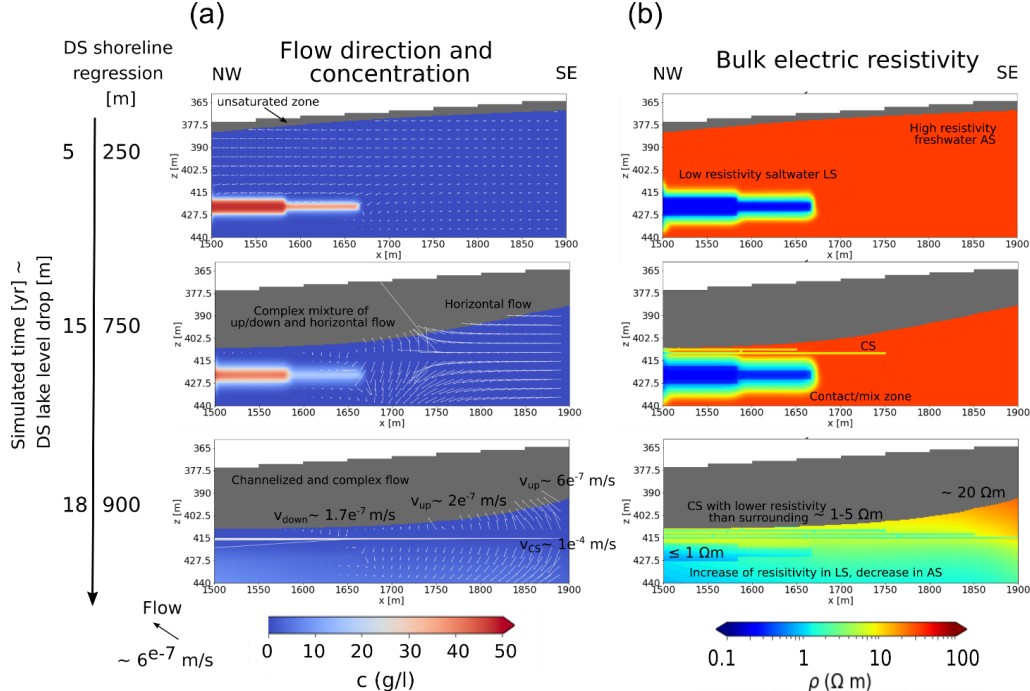

**Figure 16: Close-up views of the model alluvium/mud contact area. (a) Flow direction, typical flow values and salt concentration. (b) Simulated bulk electric resistivity. LS stands for lacustrine sediments, AS are the alluvial fan sediments and CS is the conduit system.**



## 5    Discussion

Here we first discuss the geophysical results and their interpretation, and we then discuss the hydrogeological modelling.  In view of these aspects and the insights into the evolution of surface geomorphology in the study area as derived from the remote sensing data, we then propose a refined conceptual model for the hydrogeological

evolution at Ghor Al-Haditha sinkhole area.

### 5.1    Interpretation of geophysics in the context of hydrological evolution

In terms of interpreting ERT results, it is important to note that it is generally difficult to distinguish the electrical response derived from geological properties and that derived from porewater properties without well data. Apparent resistivity depends only on the electric conductivity (i.e. the number of free electrons). Different materials can have

equal or similar conductivity, and so the material represented by a given apparent resistivity is always non-unique. Small variations in the portions of metal (e.g. Fe) or salt (e.g. Na, K), whether contained within in the solid or liquid phases, can result in large variations in their electrical conductivity. Additionally one has to bear in mind the the principle of equivalence in electric and electromagnetic methods (Kirsch, 2006). An inserted stack of thin highly conductive layers within a low conductive background layer structure can lead to similar apparent resistivities in

ERT as a thicker, less conductive layer. The conductance of both features would image similarly, and interpretation needs to be done carefully, on the basis of additional data, boreholes and conductivity measurements.

Consistent with former studies in the area (Alrshdan, 2012; Frumkin et al., 2011), our geophysical surveys on the alluvial fan (ERT1 and ERT2, Figure 12) indicate a general decrease in resistivity with depth, broadly stratified

into distinct regions. On the basis of logs of sinkhole walls nearby (Sawarieh et al., 2000; Taqieddin et al., 2000), the uppermost region at 0 – 10 m depth with middle range resistivities (30 – 100 Ωm), likely represents a well irrigated topsoil overlying alluvium with a high silt and/or clay content. This grades downward into a highly resistive (100 − 500 Ωm) region that lies at 20 – 45 m depth. The basal surface of this layer has a 'hummocky', uneven geometry, particularly in ERT1 (Figure 12c), which may represent initial sediment geometry, typical

different saturation grades and/or salinization (e.g. Farzamian et al., 2019a, 2019b; Gonçalves et al., 2017). This layer correlates with the sands and gravels logged in BH1 and BH2 by El-Isa et al. (1995). Below this, a less resistive (10 – 100 Ωm) layer extends to at least 70 m depth. VES data and models from 1994 (El-Isa et al., 1995), constrained partly by the borehole BH1, suggested a locally complex distribution of dry alluvium and alluvium with fresh, brackish, or saline water. If we consider the resistivity ranges ascribed to different groundwater

conditions by El-Isa et al. (1995) to be correct, then we would interpret the highly resistive region at 10 - 40 m depth to be equivalent to dry alluvial fan deposits and the lower more conductive region to represent the presence of groundwater. At the base of ERT2, there is a small area of low (2 − 5 Ωm) resistivity, which may represent brackish groundwater within alluvium or salty mud.

Other studies at the DS coastal area have determined resistivity values for brine saturated materials between 0.25 and 0.4 Ωm for lacustrine sediments, and between 0.45 and 5.8 Ωm for alluvial fan sediments (Ezersky and Frumkin, 2017; Yechieli et al., 2001). The groundwater resistivity has been estimated by the same two studies in the Dead Sea coastal area to lie between 0.247 and 0.765 Ωm. From our numerical modelling, highly conductive



parts (with resistivities lower than 5 $\Omega m$) may relate to either brine to saline water-saturated alluvium and or brine to saline water-saturated lacustrine sediments, and hence a geological distinction based only on the ERT surveys alone is impossible. More specifically, the conductive feature in ERT line 1 below the uvala (Figure 12c) may equally be a patch of water-saturated salty mud or salty alluvium; a distinction is not possible. However, the

projected boreholes BH1 and BH2 helped to identify the major lithological units. This information and a clear transition between resistive and conductive areas in the profiles and in the models offer indications of the fresh-saline interface rather than lithological boundaries.

Overall, these ERT results for profiles 1 and 2 indicate an apparent decline of the level of the water table in the

study area relative to the levels found by El-Isa et al. (1995) almost 24 years previously. The GPR profiles 1 and 2 hereby help to constrain the limit between saturated and unsaturated ground of 30-38 m below surface in October 2014. This limit, seen as a clear reflection in GPR, lies near the top of the less resistive layer estimated by ERT (10 – 100 Ωm) taken 4 years later, corresponding to the sand layer in the borehole logs of El-Isa et al., 1995. If we take the known water table depth (recorded in January 1995) in BH1 of 20.5 m (El-Isa et al., 1995, p.83-84) and

we estimate the water table depth in October 2018 from ERT to be 35-45 m, then we obtain a decrease in water table depth for this area of 15 – 25 m at a rate of 0.65 – 1.1 m yr⁻¹. This rate of water table fall is similar to the average rate of 0.75 m yr⁻¹ observed at the Mazra'a pumping station between 1960 - 2010 (US Geological Survey et al., 2013) and is in line with the rate of base level fall. Declines of groundwater levels at rates of ~ 0.25 m yr⁻¹ and ~ 0.9 $m\ yr^{-1}$ are reported from boreholes respectively at Tureibe (1985 - 2012) and Ein Gedi (2000 - 2014)

on the western shore of the Dead Sea (Abelson et al., 2017).

The combined results from ERT and SP potentially reveal a complex water-flow system via conduits from the edge of the main uvala toward the vicinity of CM5, the main active spring at the border of the alluvium and lacustrine sediments. Shallow subsurface channels can be resolved and are indicated by lower resistivity than the surrounding,

deformed alluvial fan deposits. One instance occurs at the edge of the uvala-like depression and below the damaged road on profile ERT1 (Figure 12c; Sec. 4.2.4). Others are found in association with the SP anomalies upslope of the stream channel head, near the interpreted base on the alluvium and just above the underlying lacustrine deposits (Figure 10b-c). The suggestion that these low-resistivity ERT anomalies represent flow in such a conduit network is consistent with electrical conductivity measurements of the springs at the head of CM5 during the 2018 field

campaign. These measurements, between 22 – 26 mS cm⁻¹, would equate to resistivities between 1 – 0.5 Ωm. Similar resistivities were measured both at the low-resistivity anomaly at the edge of the uvala, and at around 420 mbsl elevation along ERT3, which is the level of spring resurgence at CM5. A caveat is that ERT, even with a small electrode spacing as used in this study, cannot resolve the exact geometry of metre to sub-metre diameter flow pathways at depth.

The SP data indicate a dominant vertical upwards groundwater flow near the outflow point at CM5, but also horizontal (upslope) and downward directions. This is perhaps in part fracture-controlled locally, although the patterns of the SP anomalies resemble the shape and extension of the canyon head structures from aerial view. Also, in such a hydraulic setting of local artesian springs (cf. Al-Halbouni et al., 2017), an upslope part of the flow





can develop locally at the alluvium/mud contact, as indicated by the SP results (Figure 10) and by the simulated flow pattern (Figure 16a). The dominantly upward flow features may indicate that piping plays a role in sinkhole formation in the vicinity of stream channel heads, such as the case at CM5 in 2015 (cf. video supplement).

### 5.2 Feasibility of the hydrogeological model

The nature of aquifers formed at a boundary between fresh and saline water is critically dependent on the physical and chemical contrast between saline and fresh water. Fluid flow driven by density contrasts between saline and fresh water is generally described by two mathematical approaches: (1) the 'sharp-interface approximation' (Ghyben, 1888; Herzberg, 1901; Pool and Carrera, 2011), which assumes two fluids of constant density, no mixing and a pronounced saline-fresh water interface; and (2) the density-driven approach, which accounts for mixing of
the two water types within a 'transition zone' (Croucher and O'Sullivan, 1995; Henry, 1964; Simpson and Clement, 2004). Hydrogeological modelling of groundwater flow using both approaches has been previously undertaken for sites around the Dead Sea area (Alfaro et al., 2017; Odeh et al., 2015; Sachse, 2015; Salameh and El-Naser, 2000; Shalev et al., 2006; Strey, 2014; Yechieli, 2000), and provides sophisticated understanding of the flow systems and problems related to groundwater extraction. However, density-driven modelling with consideration of the salt
concentration distribution has so far only been presented by Strey (2014), for the Darga Quaternary Dead Sea group sediments, with a considerable analysis of factors controlling model performance, and Shalev et al. (2006), with an application to salt dissolution and sinkhole formation by faults as preferred groundwater conduits.

The main contribution of the 2D hydrogeological modelling in this paper is a conceptual understanding of the
hydraulic system at the Ghor Al-Haditha sinkhole area. The goal of the modelling is not the exact reproduction of this system, which is complex, heterogeneous, and subject to strong fluctuations of the groundwater table due to flash flooding events, but rather to achieve converging results for a realistic head (water table) and salt concentration. As for any modelling, limitations exist that should be borne in mind when appraising it. Firstly, we chose a simplified initial condition of salty lacustrine sediments and non-salty alluvium that is based on limited
subsurface constraints. The Dead Sea is known for long-term and short-term fluctuations of its level, with different sedimentation, evaporation and erosion periods (e.g. Bartov, 2002; Levy et al., 2020; Neugebauer et al., 2015). The resulting depositional inter-fingering of alluvium and lacustrine sediments, a geometry known from boreholes on the western side of the Dead Sea, was thus chosen to approximate this. Secondly, we assume an initial state of a rather sharp and fixed fresh-saline boundary that only changes from the onset of regression. Due to the limitation
of computer resources, diffusion processes that would have been present for a few 1000s of years while the sea level was rising slowly or fluctuating by a few metres (Bookman et al., 2004) have not been considered. Thirdly, there is a scarcity of long-term well data measurements for validation of the simulated heads. The SE model boundary condition has been adjusted to the nearest available well measurements, and the derived water levels are informed by the 1994 VES data interpretation (El-Isa et al., 1995; Sawarieh et al., 2000), the gradient of which has
been included as a starting condition. Personal communications from local farmers confirm that some wells bearing water at approximately 15 m depth around 10 years ago and extending down to 30 m depth have since fallen dry. This magnitude of groundwater head decrease is predicted in the alluvial cover in our model. Fourthly, although in homogeneous systems groundwater flow can be assumed perpendicular to the shoreline (effectively 2D), in



complex 3D aquifers, groundwater flow directions are deviated by geological heterogeneities and hence angular (Meyer et al., 2018b, 2018a). Despite these limitations, therefore, the main results from hydrogeological modelling help to understand the observed geomorphological changes and to interpret the geophysical results.

A first main finding is that a conduit system (or other form of greatly increased hydraulic conductivity) needs to develop across the interface between alluvium and lacustrine sediments to provide a discharge of the enhanced water flow resulting from the higher head gradient. This conclusion is based on dozens of different tested models of whom only a selection is presented in the Appendix B. The karst conduits develop at the boundary between initially relatively impermeable lacustrine sediments and relatively permeable alluvium to drain the system fast

enough, and therefore should be present also in areas of similarly changing hydraulic conditions. In our model, we simulate a tiered configuration of conduits, with this geometry being tied to the sea level. Indeed, such a configuration is reported for some carbonate coastal karst aquifers under similar conditions of shoreline regression and falling regional hydrological base level (Bakalowicz, 2015; Bakalowicz et al., 2008; Fleury et al., 2007). At Ghor Al-Haditha, our surface manifestations indicate that such conduits can be generated by physical erosion of

the weak alluvial and lacustrine materials, by chemical erosion of the evaporite component of the lacustrine deposits or by both processes. These processes are not explicitly simulated in the model, but incorporation of these and the related mechanical feedbacks could be subject of future work (e.g. Romanov et al., 2020).

A second important contribution of the hydrogeological model is to aid the interpretation of the geophysical results.

We used one set of transfer equations between TDS and electric conductivity specific to the Dead Sea materials (Ezersky and Frumkin, 2017, Yechieli, 2006). The approach is based on the definition of empirical parameters, the formation factors, a classical issue in hydrogeology. However, the results are in the range of expectations from our ERT surveys and therefore we consider our approach to be suitable, despite material heterogeneities. The assumption of material mixture in the simulated conduits seems generally viable and leads to relatively low

resistivities due to the low formation factor resulting from a high void space (Ali Garba et al., 2019). In nature, erosion would enlarge these conduits, such that they may become more electrically conductive or resistive, depending on whether infilled by air or water and depending on the salt content of the water. The lower resistivity of model conduits supports our interpretation of low resistivity anomalies in the ERT profiles as water-filled conduits. Our simulation of continuous salt diffusion predicts an overall resistivity decrease due to saline water

intrusion by both diffusion (in AS, LS) and advection (in CS, Sec. 4.3) within the alluvium. The model resistivity values agree broadly with the low values given by ERT at depths > 50 m under profiles 1 and 2, and so might support an interpretation of invasion of alluvium by saline water there, although interpretative ambiguity means that these values could also correspond to salty lacustrine sediments (cf. Polom et al., 2018). More broadly, this simulated process of salt-water intrusion could also in principle explain a decrease of electric resistivity observed

in the EG19 borehole at Ein Gedi on the western side of the DS from 0.17 to 0.1 Ωm concurrent with a groundwater level fall of ~ 12 m over 10 years (Abelson et al., 2017).



### 5.3 Surface stream channels, subsidence, and relationship to subsurface conduits

Recent work at the Ze'elim fan sinkhole site on the western side of the Dead Sea has highlighted critical interactions between surface water and groundwater within the evaporite karst system (Arav et al., 2020; Avni et al., 2016; Shviro et al., 2017). This builds upon past qualitative or anecdotal links drawn between high rainfall periods and

sinkhole formation at other sites around the Dead Sea (e.g. Arkin and Gilat, 2000; Taqieddin et al., 2000). Significant rainfall events and related flash floods at Ze'elim have been shown to cause rapid sinkhole collapses and wider longer-term subsidence, with undersaturated floodwater being drained by newly formed sinkholes and then resurging, nearly saturated with salt, shoreward (downstream) of these holes. Surface water ingress thus promotes the local development of sinkholes, which in turn feed surface water down into evaporite karst conduits,

which undergo chemical erosion and destabilise and form more sinkholes in a self-accelerating manner. This interlinked development of stream channels, sinkholes, and evaporite karst conduits thus occurs in a manner akin to our understanding of classical limestone karst settings.

Our study at Ghor Al-Haditha offers a 'long view' of similar processes that complements the abovementioned

studies on the western shore of the Dead Sea. Our remote sensing and geophysical data cannot resolve the effects of individual flood events as at Ze'elim, but instead indicate a major long-term shift in the hydrogeological system that is reflected in the geomorphological development of the alluvial fan section of the Ghor Al-Haditha study area. Specifically, we observe a shift of spring-fed stream activity, as evidenced by channel incision, from the southwest to the north-east of the study area over the past 30 years with a focussing of groundwater discharge at CM5 from

2012 onward (Figure 1b, Figure 7). This broadly coincident with a shift in the area of active subsidence from the south-west to the north east also (Watson et al., 2019). These shifts have occurred despite artificial excavation and straightening of the Wadi Ibn Hamad in the 1980s to guide surface flash flood water to the southwest. Such artificial drainage has enabled flash floods to bypass most of the main area of subsidence and sinkhole formation studied here (i.e. the uvala around the former factory site). Therefore, in contrast to observations at Ze'elim, it seems that

surface water influx into that area has been minimal. Only since 2006 has surface run-off from the Wadi Mutayl shown signs of being partly directed into the uvala (Figure 7), though timing-wise this appears to be a consequence, rather than a cause, of uvala development. As at Ze'elim, the eventual ingress of surface water from Wadi Mutayl into the uvala may have contributed to its accelerated development during 2006-2012 (cf. Avni et al., 2016). Nonetheless, the observed shifts in discharge location and subsidence around the main Ghor Al-Haditha alluvial

fan must overall reflect a re-routing of groundwater flow within the alluvial deposits and any potentially underlying lacustrine deposits.

Such subsurface flow re-routing is common in karst: active conduits regularly become constricted or blocked via sediment deposition or conduit collapse (e.g. Brook and Murphy, 2017; Despain and Stock, 2005; Farrant and

Simms, 2011; Palmer, 1975; Plan et al., 2009; Simms and Hunt, 2007; Šušteršič, 2006). New conduits, often termed 'floodwater diversion conduits' (Palmer, 1991), commonly develop to bypass such blockages. Another contributing factor in the north-eastward shift in groundwater resurgence with time at Ghor Al-Haditha may be the reaction of the hydraulic potential field to the uncovering of the bathymetry of the former lake bed, which is spatio-temporally variable across the study area. This is reflected in the rate of shoreline retreat, which is rapid in the southern part





of the study area and is slowest in the vicinity of CM5, relative to the 1967 shoreline (cf. Figure 3 of Watson et al., 2019).

Our study also complements the recent works on the western shore by highlighting a similar short-term link
between rainfall events, elevated stream discharge, conduit erosion and sinkhole collapses. During the high flow event of 2015, discharge at the main spring ($s_4$) clearly emanated from a conduit within the lacustrine sediments, about 6 m below the alluvium/lacustrine interface (Figure 9e-f, cf. video supplement). This altitude level is comparable to the level of suspected conduits as indicated by the SP and ERT results from 2018. In recent studies of similar processes on the Ze'elim fan (Arav et al., 2020; Avni et al., 2016), high flow events were inferred to
cause substantial erosion and rearrangement of the conduit network. This erosion seems to be largely chemical (i.e. related to subsurface salt dissolution), as evidenced by high TDS values and Na/Cl ratios in the discharging springs downstream of the sinkholes there. In contrast, we observed relatively low electrical conductivities of the water at the resurging at CM5 in both high and low flow states (2015 and 2018 respectively), in line with historical measurements in the area. This observation and the high density of vegetation at the CM5 channel head suggests
that the salt concentrations of the groundwater here are low, thus favouring mechanical erosion as a key agent of conduit development in this particular case.

These observations therefore highlight the potential role of physical erosion and the piping mechanism for generating conduits and sinkholes within the upper few metres of alluvial fan deposits and/or lacustrine deposits,
as initially proposed earlier studies at the Dead Sea (Al-Halbouni et al., 2017; Arkin and Gilat, 2000; Taqieddin et al., 2000). Further observations supporting the possibility of conduits being sustained in the alluvium include sustained water flow at the base of sinkholes in alluvium on the Neve Zohar fan on the western side (Arkin and Gilat, 2000). Such conduits are likely generated by the washing out of fines within the alluvium, especially during high base flow following recharge of the fan by rainfall and/or floods, and they can be sustained by the cementation
of the alluvium – especially the older deposits. It is stressed that such a physical mechanism of conduit generation likely goes hand in hand with conduit development by chemical erosion (i.e. salt dissolution) occurring at deeper levels, similar to what has been demonstrated at several sites on the western shore. In this view, chemical erosion and deformation at deeper levels could help to create the initial secondary porosity within the alluvium that is then further amplified by focussing of water flow in a feedback loop similar to that proposed for salt dissolution.

The spatial extent and dimensions of any conduit network at Ghor Al-Haditha remains speculative due to a lack of direct observation; however, some basic inferences can be derived regarding such a conduit network. Firstly, the subsurface development of any kind of 'branchwork' geometry, as is often observed in limestone caves, is unlikely due to the absence of a network of interconnected subsurface fractures and of concentrated surface recharge points
to the conduit network from surface depressions. As discussed in Watson et al. (2019), we regard dissolution by surface water to be almost absent at Ghor Al-Haditha. In our study area, the medium which provides the initial hydraulic connectivity are the interpore spaces in the matrix. The intergranular nature of initial porosity, combined with the observed substantial discharge variations, suggests that the pattern of conduit development may be a 'spongework' of connected, enlarged voids and pathways (Klimchouk et al., 2000; Palmer, 1991, 2007). No direct





observations of spongework conduit zones have been made in evaporite karst areas and they are rare in the case of limestone karst. The locations where spongework has been observed are often areas of mixing of different water chemistries in rocks with high primary porosity, such as mixing of fresh water and seawater along the coastal carbonate platforms of the Bahamas, where they are termed 'flank-margin caves' (Mylroie and Carew, 1990); the

coasts of the Yucatan Peninsula in Mexico (Back et al., 1984; Smart et al., 2006); and the Prichernomorsky Basin on the north coast of the Black Sea (Klimchouk et al., 2012). In the case of the Prichernomorsky Basin, water derived from deep sources rich in sulphuric acid mixes with shallower matrix flow, causing aggressive groundwater erosion under hypogenic conditions. We suspect that deep-seated groundwater sources rich in sulphur also contribute to the groundwater resurging at the alluvium-mudflat boundary at Ghor Al-Haditha: a number of the

springs feeding the stream channels, including $s_2$ at CM5 (Figure 8) but particularly those to the north of the studied area such as that feeding CM7, have a highly sulphurous odour. It has been noted that other groundwater springs in the vicinity smelling of sulphur seem to derive from the Ram and Kurnub sandstone units (IOH and APC, 1995; Khalil, 1992), which would likewise imply a deep-seated source to these waters at Ghor Al-Haditha.

### 5.4    Conceptual model of hydrological and geomorphological processes

The culmination of our study is a refined conceptual model of the surface and subsurface hydrological processes which have developed at Ghor Al-Haditha since the onset of base-level fall in 1967, and how these processes link to the geomorphological evolution of the study area in space and time (Figure 17). The fall in base level in the study area has led to the development of a strong hydraulic gradient and an ingress of fresher groundwater at decreasing elevations within the alluvial fan deposits. We propose that this recharge is derived primarily from flow

within the Wadi Ibn Hamad and secondarily from within the smaller Wadi Mutayl. Leakages from the bedrock aquifers may also play a role, especially in the northern part of the Ghor Al-Haditha sinkhole site (north of the area shown in Figure 1b), where the prevalence of sulphurous springs suggests involvement of water from the Ram-Kurnub sandstone aquifer. We propose that groundwater flow within the alluvial fan is guided towards the Dead Sea by matrix flow initially. Buried channel systems may act as groundwater pathways due to the enhanced

permeability afforded by the unconsolidated coarse sediments deposited within them. Exactly how deep these paleochannels extend into the subsurface is uncertain and likely highly variable, though the one imaged by seismic line 6 extends to depths of at least 120 m below the present land surface (Figure 13).

Upon encountering the lacustrine mud and evaporite deposits in the subsurface nearer the Dead Sea shoreline, there

is a transition to more concentrated and complex groundwater flow within a network of conduits that is developed by both dissolution and physical erosion. Water flowing within the shallowest conduits is brackish in nature, as indicated by its enhanced electrical conductivity as imaged by ERT (Figure 12c) and measured EC values at the springs feeding CM5 during the 2015 and 2018 field campaigns. The surveys presented in this paper indicate that such conduits exist between 5-30 m depth between the head of CM5 and the centre of the alluvial fan, though it is

possible that the conduit network extends to greater depths than this. Erosion within this conduit network has produced a widespread area of surface subsidence which we term an uvala (see Watson et al., 2019 for a full discussion of the processes governing uvala and sinkhole formation at Ghor Al-Haditha). These conduits direct flow into surface stream channels via springs of spatially and temporally variable discharge. The stream channels


grow during periods of activity of these springs and cease to grow when that section of the conduit network has become inactive. Spatio-temporally, the activity of surface streams fed by groundwater has shown a northeasterly shift over time (Figure 1b, Figure 7), which is likely to reflect changes within the conduit network.

Additionally, our modelling results indicate that a complex fresh-saline water interface builds up over different levels. The shallow subsurface is dominated by 'advection' transport processes within the matrix and conduit network, while the late initiation of saline water intrusion only after 17 years (or 850 m) of DS regression in our model (bottom row of Figure 15, Sec. 4.3) suggests the dominant process occurring at depth is slower diffusion across a broad 'interface zone' between saline Dead Sea water and the fresh and brackish water within the matrix

and conduit network. These saltwater intrusions of resistivities $\leq 1\ \Omega m$ may extend remarkably more towards inland due to the high salt concentration of Dead Sea water, in line with magnetotelluric studies of the deep brine system from Meqbel et al. (2013).

To help elucidate the true nature of the subsurface hydrological system in the study area and its links to surface

geomorphology, future work should include gathering more comprehensive information on the subsurface lithology and continuous discharge data for the springs and channels. An understanding of the temporal characteristics of discharge at the springs feeding these surface channels is critical to developing an improved comprehension of the links between subsurface channelization and surface collapse at Ghor Al-Haditha. Further borehole drilling would likewise provide the opportunity to improve ground truth of the geophysical surveys undertaken in the study area

and to improve our understanding of the nature of erosion in the subsurface.





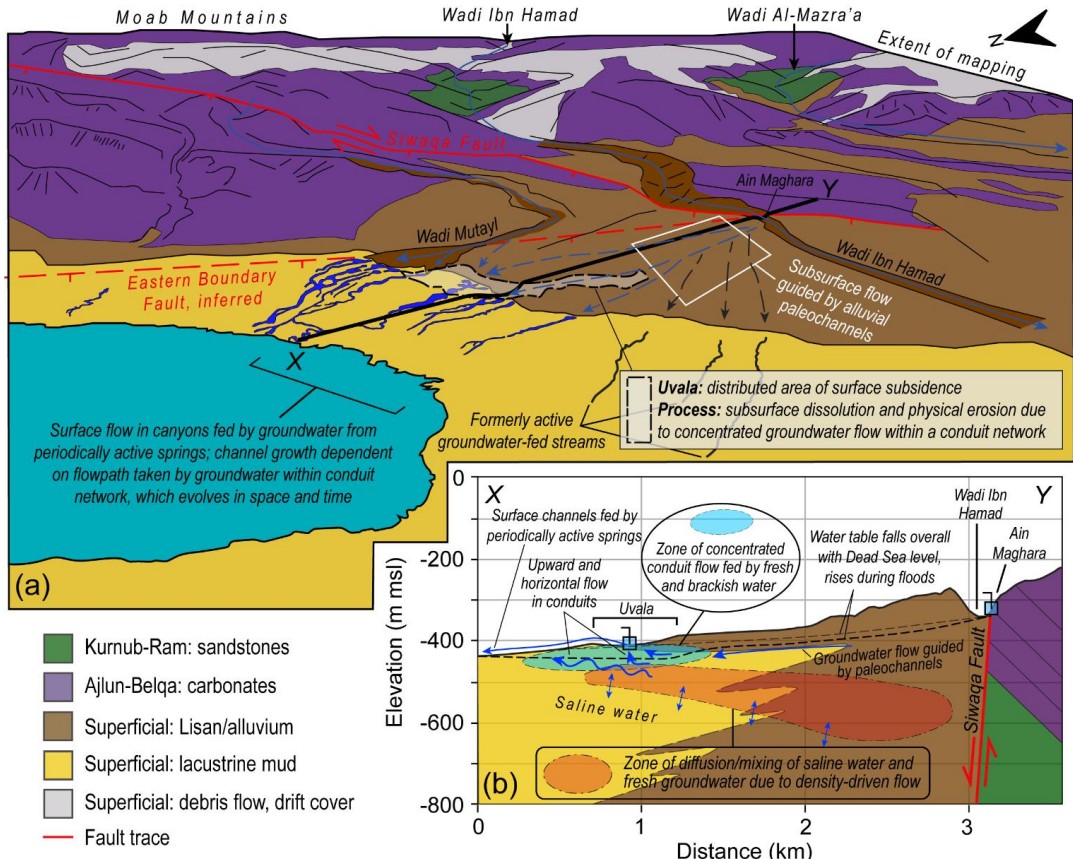

**Figure 17: Conceptual model of the hydrogeological system at Ghor Al-Haditha. (a) Oblique aerial view sketch map (altitude of the eye is around 600 m) showing the geological units (as outlined in Sec. 2, based on Khalil, 1992), structural geology, catchments of the major wadis and areas of surface flow observed in channels and inferred subsurface flow in buried paleochannels (in the alluvial fan deposits) and in a conduit network (below the uvala). Line X-Y is the location of the cross-section in part (b), which shows the hydraulic structure of the subsurface as inferred from our geophysical studies and the hydrogeological modelling.**

## 6    Summary and conclusions

This study supports the hypothesis that the subsurface hydrological flow regime at Ghor Al-Haditha contains elements of diffuse (matrix) flow and concentrated, heterogeneous conduit flow. Concentrated subsurface flow

10   appears to occur across the study area at depths ranging between 5 – 30 m deep, bringing fresher groundwater to the constantly-retreating Dead Sea shoreline.

At the former shoreline, the distribution of surface water flow is directed by subsurface conduit flow. During the studied period, surface stream channels (fed by springs occurring at the contact between alluvial gravels and

15   lacustrine sediments of the former lakebed) dried up as subsurface flow re-routed to converge upon a new network of springs draining into a now-dominant central channel (CM5) which formed in 2012 at the centre of a major karstic depression hosting hundreds of sinkholes. ERT and SP data from the head of this channel resolve shallow subsurface anomalies inferred to represent a complex pattern of multidirectional water flow.



Further inland, new ERT surveys reveal features which we interpret to represent concentrated subsurface water flow at around 30 m depth directly beneath the outer limits of the major karstic depression. GPR and ERT data give constraints on the groundwater table which has fallen by 15 - 25 m within 23 years. Still further inland close to the major wadi which drains the mountains to the east, shear wave seismic reflection surveys reveal buried paleo-

channel deposits up to 120 m deep. We hypothesise that the paleo-channel now acts as a pathway for groundwater flow within the alluvial superficial aquifer towards the distal parts of the Wadi Ibn Hamad fan delta in the locality of the now-dominant central channel CM5, where there are finer (and potentially soluble) materials. This enhanced groundwater flow ultimately promotes the formation of (and further flow concentration into) a network of conduits made by piping and/or dissolution.

Numerical simulations also suggest that conduit flow helps to maintain the interface between fresh and saline water at the shoreline as the Dead Sea level falls over time. Realistic simulations (as compared to studies on the Western shore and elsewhere) of the evolution of salt concentration and groundwater head under density-driven flow can only be achieved by simulating an increasing number of conduits of high hydraulic conductivity in the model space

as the Dead Sea level (and consequently the groundwater level) falls over time after starting the model run with diffuse flow alone. Furthermore, simulated electric resistivities are in good agreement with results from ERT.

These combined lines of evidence shed light on the increasing influence of underground conduit flow across the study area in space and time as base level fall has proceeded.  This process leads to a complex flow pattern at

hydrogeological boundaries and plays a key role in the formation and spatio-temporal distribution of canyons and surface depressions that have formed extensively across the study area, confirming the results of previous studies, and may also act to maintain density-driven flow as the hydrological base level of the Dead Sea falls over time.

**Data availability:**

All geophysical data are available on request from the authors. Free satellite images (Corona) and photogrammetric survey raw images, DSMs and orthophotos are available upon consultation with the authors. Geological Map 1:50,000 Ar Rabba: 675 available at discretion of MEMR.

**Video supplement:**

Three videos of the karstic spring $s_4$ formation at canyon CM5 and associated upstream canyon growth by retrograde erosion are provided in the online supplement.

**Author contributions:**

DAH led the conception and writing of the manuscript. DAH, RAW and EPH designed and planned the field

experiment and largely assembled the manuscript. DAH and RAW created and assembled the figures. DAH analysed the ERT, SP and GPR data; FDS analysed the SP data; RAW analysed RS data; UP analysed the seismic data; XC analysed the GPR data. RM supervised and verified the numerical simulations. RAW and HAR participated in field data collection. DAH conceived the original idea. CMK and TD supervised the project. All authors contributed to improvement and editing of the manuscript.



**Competing interests:** The authors declare that they have no conflict of interests.

**Acknowledgments**

We would like to acknowledge our colleagues from the Jordanian Ministry of Energy and Mineral Resources, particularly Nedal Atteyat for the logistical and organisational support. We acknowledge the tireless support during fieldwork in 2018 by Cécile Blanchet, Jamal, Anas Maaitah, Emad and Rshud. For technical discussions we thank

Mohammad Farzamian and Joana Ribeiro Alves from IDL. We are grateful to Jan Igel from LIAG for analysis of the ground samples, Oliver Ritter from GFZ for providing equipment and discussions, Erich Lippmann for technical support and Damien Closson for providing photos. Furthermore, we thank colleagues from the Institute Dom Luís from Lisbon, Portugal, the UFZ in Halle, Germany, and the Technical University of Berlin, Germany, for logistical and equipment support of the various field campaigns. Thanks go to the GFZ expedition fund for financial support

of this work. We thank the European Space Agency for funding the acquisition of the Pleiades 2018 satellite image via their Third Party Missions Data Service Request scheme (Proposal ID: 40571). Much of the mapping, geographical analysis and field work was done as part of RAW's masters research project, which was supervised by EPH and funded by Geological Survey Ireland under a GSI Short Call grant to EPH (Contract Number: 2017-sc-002).

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

**Appendix A**

**Detailed background of the geophysical methods and additional results**

We here present details on the different geophysical methods and additional supportive results.

**Self-Potential:**

The following background on Self-Potential is based mainly on comprehensive works from (Jardani et al., 2007;
Jouniaux and Bordes, 2012; Jouniaux and Ishido, 2012; Richards et al., 2010; Vichabian and Morgan, 2002). For more details, the reader is referred to these authors and references within.

Clay minerals carry negative superficial electrical charges due to crystallographic defects. Attracted positive charges (cations) stay therefore at the surface of clay mineral aggregates. In the contact zone between groundwater
and soil, a solution of such neutral assemblies of ions develops, but there exist different zones where the cations largely outnumber the anions attracted by the charge difference. Physically it can be described following the Stern-Gouy-Chapman (Revil and Linde, 2006) model which divides the EDL into a high cation concentration Stern-layer, a diffuse double layer and a constant concentration layer. A zeta potential, or electrokinetic potential, is defined as the electrical potential between the solid mineral as a shear plane and a zero-potential surface (usually the liquid).
This definition leads to negative zeta-potential for the movement of positive charges and forms the physical background for the calculation of the self- or streaming potential. We derive the governing main equations for streaming potential calculation in the following. The coupled flow equations for saturated case are after (Sill, 1983):

$$J_e = -\sigma_0 \, \nabla V - L_{ek} \, \nabla P$$
$$J_f = -L_{ek} \, \nabla V - k_0/\eta_f \, \nabla P$$

(A1)

$J_f$ is hereby the primary flow of matter (e.g. Darcy flow of water) and the effects of the coupled secondary flow $J_e$ are usually considered as negligible. However, here $J_e$ is the point of interest. V is the streaming potential for
constant temperature and no concentration gradients measured as a difference between the potentials in the liquid and in the grain. $L_{ek}$ is the electrokinetic coupling coefficient following Onsager´s reciprocal relation in steady state conditions. $\eta_f$ is the dynamic viscosity of the fluid, $\sigma_0$ its' specific conductivity, $k_0$ the hydraulic conductivity and $\nabla P$ the pressure gradient.

Conservation of current ($\nabla \cdot J_e = 0$), the electric convection current is compensated by conduction current, leads to
Poisson´s equation with a source term:

$$\nabla \cdot (\sigma_0 \, \nabla V) = \nabla \cdot (L_{ek} \nabla P)$$

(A2)

For a homogeneous medium:

$$\nabla^2 V = C \, \nabla^2 P$$

(A3)

With C as the streaming potential coupling coefficient defined when $J_e = 0$:





$$C = - L_{ek}/\sigma_0 = \partial V/\partial P \qquad \textbf{(A4)}$$

The streaming potential coupling coefficient for unidirectional flow of a liquid (anode) between two electrodes (cathodes) derived from the classical electroosmotic model (Overbeek, 1952) can be expressed as:

$$C = \partial V/\partial P = \varepsilon_f \, \zeta/\eta_f \, \sigma_{eff} \qquad \textbf{(A5)}$$

With $\sigma_{eff} = F\sigma_0$ as the effective conductivity and $\varepsilon_f$ the permittivity of the fluid. $\zeta$ is the zeta potential inside the electrical double layer. This Helmholtz-Smoluchowksi equation is valid for a small double layer thickness (Debye-

5   length) in comparison with the pores and capillaries, and, at constant viscosity and permittivity if fluid conductivity dominates. Typical SP gradients of groundwater flux along an axis x are around $dV/dx = 0.2$ mV m$^{-1}$. For karst aquifers this equation holds for any capillary and the SP signal comes from capillaries and matrix, a highly amplified signal is created if e.g. sand is in the fracture. The signal is a combination of topographic regional-scale flow and e.g. sinkhole downward seepage.

The detailed results of the inversion of five SP profiles extracted from the SP array are summarized in the following table, the result for SP3 is presented in Figure A1. They represent the model parameters of geometry and electric charge of the modelled thin sheets (called strips) that extend infinitely horizontally perpendicular to each profile. The shape factor q approaches zero for such modelled horizontal strip like structure. The dipole moment for

15   inversion has been constrained to positive values only and relates the current density J to resistivity for a thin polarized strip like structure to $K = J\pi \, {}^{\rho_A}/_2$. For more details on SP modelling and inversion see (Biswas, 2017; Monteiro Dos Santos, 2010; Murthy and Haricharan, 1985).

**Table A1: Inversion results for SP modelling with the PSO method of Monteiro Dos Santos (2010).**

| Profile | Body | Distance x [m] or y [m] | Depth z [m] | Half-width a [m] | Inclination angle $\alpha_i$ [°] | Electric current dipole moment K [A*m*s] |
|---|---|---|---|---|---|---|
| SP1 | 1 | 22.20 | 5.1 | 3.2 | -110.0 | 8.6 |
| | 2 | 50.00 | 5.0 | 2.0 | -180.0 | 1.43 |
| | 3 | 86.78 | 15.0 | 16.0 | -138.99 | 18.88 |
| SP2 | 1 | 25.83 | 20.0 | 15.0 | -63.83 | 13.97 |
| | 2 | 53.82 | 8.9 | 5.0 | -120.0 | 19.73 |
| SP3 | 1 | 1.48 | 2.3 | 1.2 | 71.75 | 5.9 |
| | 2 | 27.39 | 2.0 | 2.4 | 0 | 9.49 |
| SP4 | 1 | 15.55 | 4.3 | 2.3 | 0.38 | 19.24 |
| | 2 | 25.00 | 7.0 | 2.2 | -84.84 | 8.89 |
| SP5 | 1 | 35.68 | 5.6 | 2.4 | -0.41 | 6.16 |
| | 2 | 10.04 | 11.5 | 1.0 | 90.0 | 11.24 |





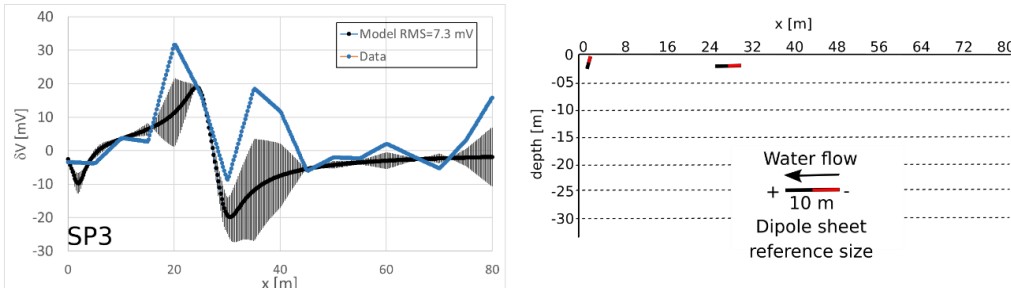

**Figure A1: SP data and inversion results for profile SP3.**

**Electric Resistivity Tomography:**

Measuring the apparent resistivity to $\rho_A = K \, {}^U\!/_I$ by electric current $I$ and voltage $U$ needs the estimation of geometric factors $K$ for each survey layout. Generally, these are calculated by the distance differences between two electrodes (A, B, M, N):

$$K = 2\pi \, [\left(\frac{1}{AM} - BM\right) - (\frac{1}{AN} - \frac{1}{BN})]^{-1} \qquad \text{(A6)}$$

For the Schlumberger configuration we get:

$$K = \pi \, (\frac{a^2}{b} - \frac{b}{4}) \qquad \text{(A7)}$$

For Wenner Alpha configuration we get:

$$K = 2\pi \, a \qquad \text{(A8)}$$

Estimation of the penetration depth however is difficult subject to the concept of conductance vs. thickness of a layer, as both can influence the distribution in the same way. The integrated conductivity can be derived by inversion methods as described by Loke et al. (2018), and in the corresponding part of the manuscript (Sec. 3.3.1).

**Ground penetrating radar and sample laboratory tests:**

Water content may lead to strong damping of the electromagnetic signals by direct current losses, preventing sufficient depth penetration of GPR signals. Therefore, laboratory measurements have been performed to determine the properties of alluvial material samples from the field site. Figure A2 contains the results for electromagnetic laboratory tests on the samples with a frequency of $f = 100$ MHz. Samples had a high iron mineral and salt

content, and, together with gravimetric soil moisture θ between 5 (in situ) and 20 %, representative for this irrigated area, the conductivity $\sigma_{dc} = 1/\rho_A$ lies between 2.7 mS m⁻¹ and 0.5 S m⁻¹. The relative dielectric permittivity ε lies between 4 and 24, depending also on the moisture content. This results in an attenuation of 10-100 dB m⁻¹ for 100 MHz and the penetration depth δ, or skin depth, for a 100 MHz antenna in a homogeneous half space can be approximated to $\delta = \sqrt{\frac{2\rho_A}{\omega \mu}}$ (Jol, 2009) with $\mu = \mu_r \mu_0 = {\sim}1 * 12.57 * 10^{-7}$ N A⁻² as the magnetic permeability

and $\omega = 2\pi f$ as the angular frequency. This results in $\delta \sim 35$ cm only for a wet soil ($\sigma_{dc} = 0.45$ S m⁻¹). For dry soil ($\sigma_{dc} = 0.0027$ S m⁻¹), the penetration depth would be high (~ 970 m) if no other attenuation effects were to be considered.

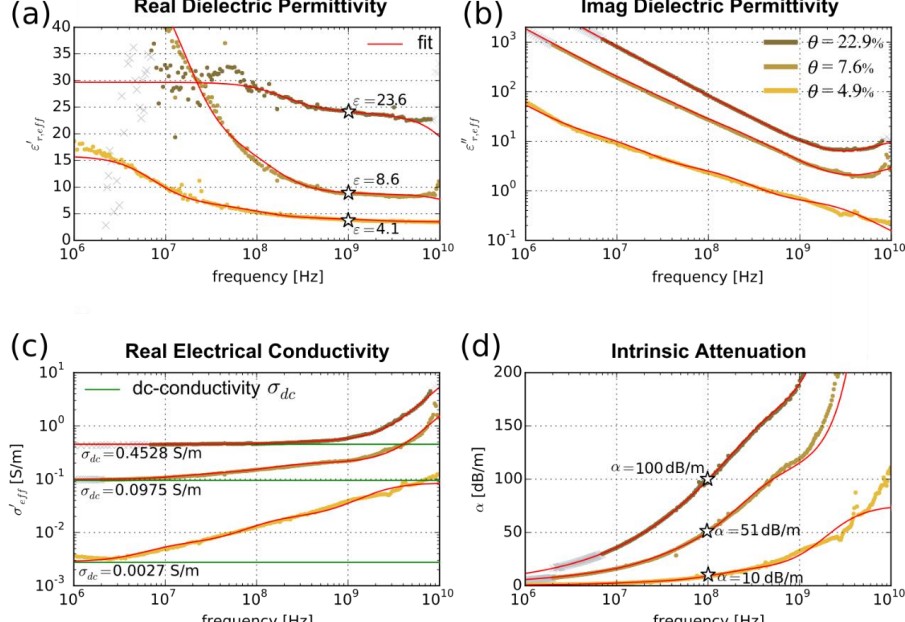

**Figure A2: Results of electromagnetic laboratory tests on alluvium samples for three different moistures θ. (a) Real part of dielectric permittivity ε. (b) Imaginary part of dielectric permittivity ε. (c) Direct-current conductivity $\sigma_{dc}$. (d) Attenuation α. The results show that attenuation in the moist alluvial material is high due to high DC conductivity. For dry alluvium damping is comparably low.**

The frequency has an impact on penetration depth and resolution of the GPR surveys (Jol, 2009). A low frequency antenna was preferred in this survey because the target, void structures and groundwater table are expected to be at more than 10 m depth, although one has to accept the low horizontal resolution of $\Delta l = \sqrt{\frac{v_{em}r}{2f_c}}$ with r as the radial distance from the sender. For 50 MHz and 0.15 m ns$^{-1}$ velocity the minimum horizontal extension of an object that can be resolved in 10 m depth is 3.87 m. For 100 MHz it is 2.7 m.

## Appendix B

### Hydrogeological model parameters, alternative models and resistivity calculation

The hydrogeological modelling procedure as described in Sec. 3.3 requires the definition of further important parameters such as effective porosity and concentration distribution, initial conditions and fixed head and concentration locations (Figure B1).

Alternative models for a convergent steady state solution have been tried and are presented in Figure B2. There are no channels installed in these models, rather only the hydraulic conductivities of lacustrine sediments are varied. Note that in model 1 the water table does not follow a realistic curve. In model 2 we achieve a more realistic shape of the water table, but the overall magnitude of the hydraulic conductivity of LM had to be increased to at least one order of magnitude higher than AS material, an unrealistic scenario for the whole lacustrine material. Hence, these models were depreciated in favour of the model presented in Sec. 4.3 with local channels of high permeability.



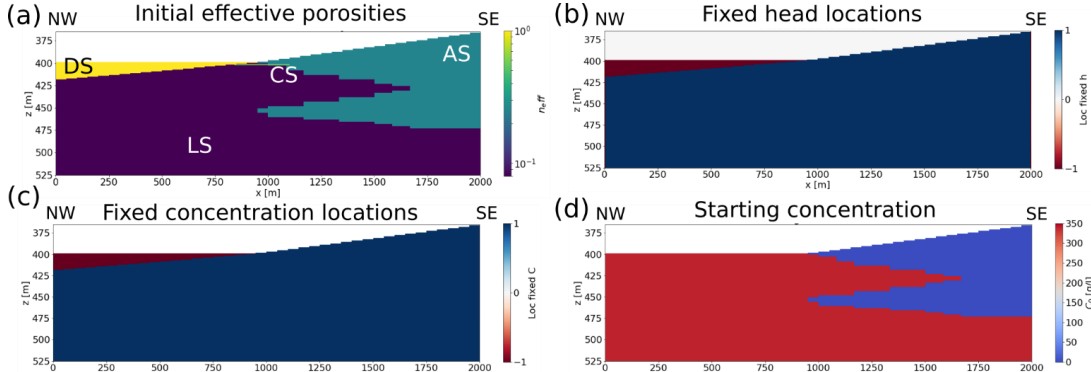

**Figure B1: Hydrogeological model parameters: (a)** Initial effective porosities for all materials. **(b)** Location of fixed head (-1), otherwise free head (1). Note that boundary conditions are also fixed but not visible due to the small cell width. **(c)** Location of fixed concentration (-1), otherwise free concentration (1). **(d)** Initial salt concentration distribution. Dead Sea and lacustrine material are valued by high values (340 g l⁻¹), the alluvial sediments by low values (0.3 g l⁻¹). LS stands for lacustrine sediments; AS are the alluvium sediments; DS is the Dead Sea and CS stands for the conduit system.

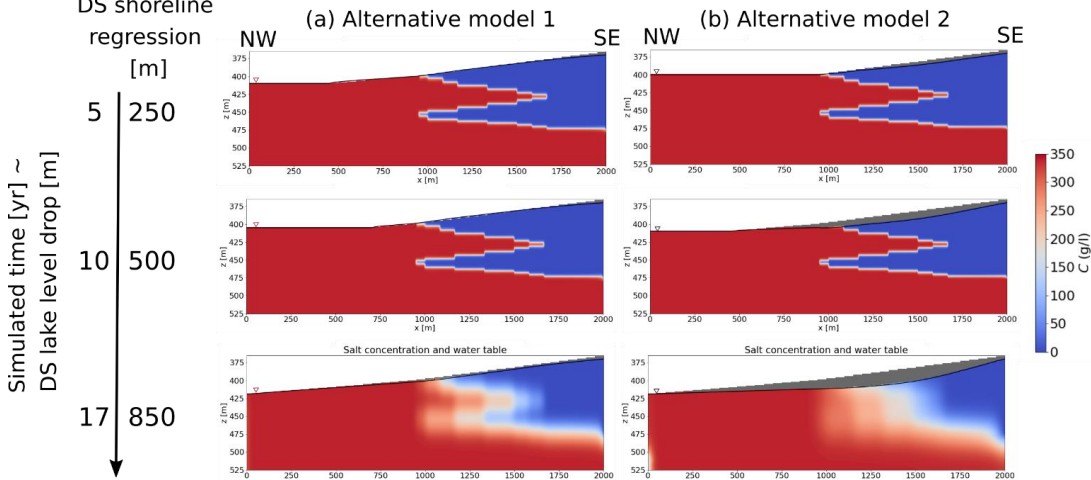

**Figure B2: Alternative (depreciated) hydrogeological model results for concentration and water table. (a)** Model 1 with horizontal and vertical hydraulic conductivity of LS of the same value as AS ($k_h = 1e\text{-}5 \text{ m s}^{-1}$, $k_v = 1e\text{-}6 \text{ m s}^{-1}$). The water table behaves in an unrealistic (totally linear) way. **(b)** Model 2 with horizontal and vertical hydraulic conductivity of LS one order of magnitude higher ($k_h = 1e\text{-}4 \text{ m s}^{-1}$, $k_v = 1e\text{-}5 \text{ m s}^{-1}$) than of AS. This high hydraulic conductivity for mud is unrealistic for the initial aquiclude conditions.

To calculate the resistivity from salinity simulations, one can alternatively to Sec. 3.3.2 use the direct bulk soil resistivity-salinity relationships from Ezersky and Frumkin (2017):

$$TDS = \rho_A^{-3.35} \times 15.5 \text{ for high concentrations (TDS} \geq 131 \text{ g l}^{-1}) \text{ in AS}$$

$$TDS = \rho_A^{-1.377} \times 42.66 \text{ for low concentrations (TDS} < 131 \text{ g l}^{-1}) \text{ in AS}$$ **(B1)**

$$TDS = \rho_A^{-2.61} \times 8.52 \text{ in LM}$$

The results differ only in the magnitude by a constant shift to lower values of roughly 30 Ωm and are not shown here.