# Peer review of "Dynamics of hydrological and geomorphological processes in evaporite karst at the eastern Dead Sea – a multidisciplinary study"

_Hydrology and Earth System Sciences, 2021_

## Referee Comment (RC2)

[referee-annotated manuscript omitted]

---

## Author Comment (AC2)

***Answer to comments of Reviewer 1***

We would like to thank reviewer 1 for the constructive and fruitful comments, suggestions and improvement. Our answers and changes follow below for each mentioned point.

- *Could you please present the results of regional and local geological summary that you used for geophysical data interpretation? This would help readers get a sense of how you calibrated geophysical data.*

*In Sec. 2 we presented quite a complete overview of the geological and hydrogeological information obtained through various surveys in the last decades. The combined findings are used for interpretation of the geophysical data, and so the reader is now referred back to Sec. 2 for comparison.*

- *Page 5: The Figure 1 (b). It is difficult to identify Ground penetrating Radar and S-Wave reflection seismic profiles.*

*Figure 1 has been updated to improve the clarity and legibility of the labels.*

- *Page 6: Figure 2. Could you please indicate on the field photos the limits of alluvial fans, halite cover, silt-clayey marl deposits, and alluvial deposits?*

*We now indicated the limit between mud/saltflat and alluvial fans by a dashed white line in the upper left figure (part a) and adapted the figure caption accordingly. However, mud/saltflat refers to a large area in which halite cover and and clay/marl materials are interleaved, so distinguishing their limits is not practical. Themore detailed figures to the right serves to highligt the interleaving of the lacustrine deposits.*

- *Page 8: Table 1. Could you please provide the Spring Id with logical numeration? Or provide the missing information for (4,5,6) Spring Id.*

*The water sampling data presented in Table 1 is not our own, but is that collected by Sawarieh et al. (2000) (pg 29, Table 4-1). Therefore, to aid comparison and maintain consistency between the studies, we have used their numbering system for the water sampling data. Sawarieh et al. sampled sinkholes and springs in their work; since we are not interested in the sinkhole data, we omitted these from our table, hence the „missing" sample numbers. Figure 1 and the caption to Table 1 have been updated to make the source of these spring data clearer and to explain the origin of the numbering.*

- *Page 10: line 28: Pansharpening pre-processing: What were the results? Did you obtain the same spatial resolution for all the satellite images you used? These could help readers understand the smallest fluvial and karst features you were able to identify and extract on remote sensing data.*

*The satellite imagery pre-processing steps and spatial resolutions were detailed in a previous publication using the same data (Watson et al., 2019). Therefore, we have decided not to replicate the information here (especially in light of comments from reviewer 2 regarding the length of the manuscript). For details on the satellite imagery pre-processing, readers should refer to Section 3 and Table 1 of Watson et al. (2019).*

- *Page 10: lines 32-34: What were the band combinations you chose for aerial and satellite images on which you manually digitalize the fluvial and karst features? Which bands did you choose? What was the base on which you chose these bands? Did you notice any band combination which highlights better the fluvial and karst features? Please provide more details.*

*In the case of all of the high-resolution sensors which provided data for this study, only 4 multispectral bands are available: red, green, blue and near infra-red. All analysis was performed using 'natural colour' (R-G-B) imagery and all data is presented in this format to show the study area as it would appear in the field. Again details are available in the paper by Watson el al (2019), and the reader is referred to that work.*

- *Page 11: Lines 17-19: Please provide spatial distance between each SP point measurements.*

*Thanks for pointing this out. We indicated the spatial distance of 10 m now in the corresponding part.*

- *Page 16: Figure 6: Please indicate the limits of CS (with a box).*

*This is done now with a dashed line box in Fig. 5 (note the shift of figure numbers after Fig. 4).*

- *Pages 20-21: Figure 7-8: Could you please provide the name of satellite and the band combination for aerial and satellite imagery presented on the left column of these Figures?*

*Names of the satellite/survey have been provided in the figure caption. For band information, please refer to the response to the previous comment.*

*With best regards,*

*Djamil Al-Halbouni on behalf of all co-authors.*

---

## Author Comment (AC3)

***Answers to comments of Reviewer 2***

We would like to thank reviewer 2 for the constructive comments and suggestions for improvement. Our answers and changes follow below for each mentioned point. Additionally, typos, repetitions & grammatical mistakes throughout the whole document were corrected according to the reviewer's annotations.

- *The main point is the length of the manuscript, which seems to be excessive, and even with many repetitions of terms in the same paragraphs. These repetitions should be avoided, by using other terms and making efforts to vary the text. I suggest reducing the text especially as regards the explanation of the different geophysical techniques, which are well known in the scientific literature, and do not need lengthy descriptions. At the same time, many details about the methods used for the study could be moved in the supplementary materials, in order to not make the reading too heavy.*

We have adopted the reviewer's suggestion and moved a large part of the method description into Appendix A, including the former Figure no. 4. This way the description is more compact, but still detailed enough to assist readers with potentially low knowledge of geophysical methods. For this purpose we explicitly left the details on field application of the methods in the main body of the text.

We also removed repetitions as suggested in the supplementary pdf file of the reviewer. Note that due to the moving of former Figure 4, the subsequent figure numbers have changed, as have the figure numbers also in Appendix A.

- *Hydrogeological modelling: this part is very interesting, but my feeling is that more details should be provided as concerns some of the constraints of the model.*

We included a revised hydrogeological model (Sections 3.3.2 and 4.3) that is spatially extended and better contrained to the available data. The results are largely unchanged but the extension to 4 km x 400 m of the model enables a better comparison with the local geology. The revised model includes a longer temporal simulation by adjusting the Dead Sea regression to estimations from Watson et al 2019 over the last 50 years and the higher resolution provides even more accurate information about the fresh-saline water interface. Accordingly, figures 6, 15, 16, B1 and B2 are adapted to the extended model without significantly changing the scientific content & conclusions.

- *In particular, it is stated (p16 l 11) that "Vertical hydraulic conductivities are assumed to be 10 times less the horizontal ones, to represent anisotropy imparted from sedimentological layering". This assumption makes bedding as the main feature favouring conductivity, limiting very much the likely role of tectonics, which I believe is significant as well, as also documented in many articles. It would be therefore necessary to present some evidence to support this assumption, which should be presented in the section about hydrogeological modelling, and also in the conclusions.*

*The precise role of tectonics in groundwater flow in this area is unclear. On one hand several studies have used an apparent parallelism of local sinkhole alignments and regional fracture orientations as a basis for invoking vertical groundwater flow along buried faults within the*

*superfical aquifer (e.g. Shalev et al., 2006; Closson and Abou-Karaki 2009). One the other hand, other studies have argued that such a control of faults on sinkhole alignment is indirect: the faults simply controlling the lateral extent of evaportie deposuts, rather than faclitating vertical groundwater flow (Ezersky et al 2006; Watson et al 2019).*

*Even if one were to adopt the view that faults control groundwater flow significantly in the shallow subsurface of the study area, one would have to make further assumptions about whether the faults act as conduits or barriers to flow. Particularly in the case of poorly consolidated, clay-rich sediments (such as in the study area) smearing of clay along fault surfaces may reduce, rather than increase, their permeability and this may cause them to act as barriers to both vertical (i.e along fault) flow and horizontal (i.e. across fault) flow. For a review of this topic, see the paper by Vrolijk et al. ( 2016, J. Struct. Geol.). Given that the geometry, extent and nature of putative fault zones within the study area are not independently constrained, we therefore choose the conservative assumption that sedimentological layering (which is observed near surface though also poorly constrained at depth) is the dominant lithological influence on groundwater flow.*

- *When quoting more than one reference, the list should be organized following the chronological order. In many places throughout the article this has not been followed. Please format the text strictly following the journal guidelines.*

This has been improved throughout the whole manuscript.

- *As regards references, I suggest here some additional references about karst evaporites and sinkhole development, which could be useful especially in the introductory part of the article, also to put it in a broader international context of evaporite karst:*

*Bruthans J., Asadi N., Filippi M., Vilhelm Z. & Zare M., 2008.Erosion rates of salt diapirs surfaces: An important factor for development of morphology of salt diapirs and environmental consequences (Zagros Mts., SE Iran). Environmental Geology, 53 (5): 1091-1098.*

*Bruthans J., Filippi M., Zare M., Churáĕ ková Z., Asadi N., Fuchs M. & Adamoviĕ J., 2010. Evolution of salt diapir and karst morphology during the last glacial cycle: effects of sea-level oscillation, diapir and regional uplift, and erosion (Persian Gulf, Iran). Geomorphology, 121: 291-304.*

*De Waele J., Piccini L., Columbu A., Madonia G., Vattano M., Calligaris C., D'Angeli I.M., Parise M., Chiesi M., Sivelli M., Vigna B., Zini L., Chiarini V., Sauro F., Drysdale R. and Forti P., 2017. Evaporite karst in Italy: a review. International Journal of Speleology, vol. 46 (2), p. 137-168.*

*Dreybrodt, W., 2004. Dissolution: evaporite and carbonate rocks. In: Gunn, J. (Ed.), Encyclopedia of Caves and Karst Science. Fitzroy Dearborn, New York, pp. 295–300.*

*Filippi M., Bruthans J., Palatinus L., Zare M. and Asadi N. 2011. Secondary halite deposits in the Iranian salt karst: general description and origin. International Journal of Speleology, 40 (2), 141-162.*

*Gutiérrez, F., Cooper, A.H., Johnson, K.S., 2008. Identification, prediction and mitigation of sinkhole hazards in evaporite karst areas. Environ. Geol. 53, 1007–1022.*

*Gutiérrez, F., Linares, R., Roqué, C., Zarroca, M., Rosell, J., Galve, J.P., Carbonell, D., 2012. Investigating gravitational grabens related to lateral spreading and evaporite dissolution subsidence by means of detailed mapping, trenching, and electrical resistivity tomography (Spanish Pyrenees). Lithosphere 4, 331–353.*

*Iovine G., Parise M. & Trocino A., 2010. Breakdown mechanisms in gypsum caves of southern Italy, and the related effects at the surface. Zeitschrift fur Geomorphologie, vol. 54 (suppl. 2), p. 153-178.*

Thanks for the references. We included the ones that are relevant to the background of this study.

- *P2 l39 – Add references:*
  *Closson D, Lamoreaux PE, Abou Karaki N, Al-Fugha H (2007) Karst system developed in salt layers of the Lisan Peninsula, Dead Sea, Jordan. Environ Geol 52:155–172.*

  *Parise M., Closson D., Gutierrez F. & Stevanovic Z. (2015) Anticipating and managing engineering problems in the complex karst environment. Environmental Earth Sciences, vol. 74, p. 7823-7835.*

Done.

- *P4 l6: Add reference Palmer 2007.*

Done.

- *P13 l23: Add reference*
  *Margiotta S., Negri S., Parise M. & Quarta T.A.M., 2016, Karst geosites at risk of collapse: the sinkholes at Nociglia (Apulia, SE Italy). Environmental Earth Sciences, vol. 75 (1), p. 1-10, DOI: 10.1007/s12665-015-4848-y.*

Done.
- *P13 l26 Add reference*
  *Margiotta S., Negri S., Parise M. & Valloni R., 2012, Mapping the susceptibility to sinkholes in coastal areas, based on stratigraphy, geomorphology and geophysics. Natural Hazards, vol. 62 (2), p. 657-676, DOI 10.1007/s11069-012-0100-1.*

Done.

- *P18 l 34: It would be nice to indicate the type of sinkhole mechanisms involved. At this regard, please see Gutierrez et al. (2014) and Parise (2019).*
  *Parise M., 2019, Sinkholes. In: White W.B., Culver D.C. & Pipan T. (Eds.), Encyclopedia of Caves. Academic Press, Elsevier, 3rd edition, ISBN ISBN 978-0-12-814124-3, p. 934-942.*

This part is rather related to the spatio-temporal development of sinkholes at Ghor Al-Haditha. A description of the mechanism seems out of place here. We added the type of mechanism and referred to the original publication the above mentioned general definitions of Gutierrez and Parise now in the introductory part (P. 4 l1-5).

- *P3 l36++: This point is very important, I would stress it a little more to explain to the reader the inter-mutuality among the mentioned processes.*

Indeed the content of this paragraph was interrupted by a different information on mechanical sinkhole collapse. This is now moved further below and as a consequence, the process description now is more concise and the intermutuality explained in the whole paragraph.

- *P32 l 11 & P39 l8-13: This is a relevant assumption, since the flow into karst conduits cannot typically be described as in equivalent porous medium. This strong limitation should be better highlighted to the reader. Once again, this confirms the role played by karst conduits though turbulent flow of water, which is in clear contrast with the assumption of equivalent porous medium in the hydrogeological model.*

Referring to Goldscheider and Drew (2007) there is a variety of modelling approaches to simulate karst conduit flow either by EPM, double continuum, discrete fracture networks, continuum-discontinuum approaches etc. The EPM approach is not uncommon in karst studies and successful examples for a rather general representation of groundwater flow systems include e.g. Ghasemizadeh et al. 2015. It is clear that we cannot capture irregular conduit pathways and turbulence. In our case we lack the data necessary to take a more refined approach with the modelling. However, for the purpose of a conceptual understanding, the EPM approach is suitable and computationally at low cost in comparison e.g. to a DFN approach. We now highlighted this assumption more to the reader and added a discussion point (p. 40 l1-5). In near future we also plan to use a coupled finite-discrete approach for the application to the eastern Dead Sea.

Ghasemizadeh, R., Yu, X., Butscher, C., Hellweger, F., Padilla, I., Alshawabkeh, A., and Cao, B. Y.: Equivalent porous media (EPM) simulation of groundwater hydraulics and contaminant transport in Karst aquifers, PLoS One, 10, https://doi.org/10.1371/journal.pone.0138954, 2015.

Goldscheider, N. and Drew, D.: Methods in Karst Hydrology, Internatio., edited by: Robins, N., Taylor and Francis, Wallingford, 2007.

And others.

- *P41 l1+2 and l6: This is not true. There are many caving reports of spongework-type caves explored. This should be corrected, and other caves are being documented as formed by such processes in the Mediterranean area (Ruggieri & De Waele, 2014; D'Angeli et al., 2015; Arriolabengoa et al., 2017).*

We thank the reviewer for drawing our attention to these interesting and relevant studies of spongework and flank margin caves in Italy, which we have now made reference to in the manuscript, and we agree that we may have overstated the scarcity of spongework cave passage currently mapped. However, we nevertheless maintain that available data demonstrate that spongework cave occurrences are 'uncommon' in limestone karst. For example, Palmer (1991) studied 500 caves from geological and geomorphic settings across the US and found that 'spongework' describes around 5 % of the caves (as entities) in the study but can only be applied to less than 1 % of the aggregated length of all cave passages studied. We have updated the manuscript accordingly to reflect the reviewers' comments and this response to those comments.

- *Some figures show some problems, mostly as concerns inner writings. These are often too small to be easily read, and should therefore be increased to facilitate the comprehension of the figures. E.g. Fig. 3: Writings in the inset c) are too small to be clearly read. & Fig 16: Writings to small*

Annotations on Fig. 16 are increased in size. We decided to increase the size of the figures 3 and 9 to full pages and will request the journal to keep it that way, therefore the annoations there should be clearly readible.

With best regards

Djamil Al-Halbouni on behalf of all co-authors